# Learning Efficient and Interpretable Multi-Agent Communication

**Wei Du**[1,2], **Benyu Wu**[1], **Yuqing Sun**[1], **Wei Guo**[1,2], **Yuntao Du**[1], **Zhongmin Yan**[1],
**Guoxian Yu**[1,2, *], **Lizhen Cui**[1,2,*]
[1]School of Software, Shandong University, China
[2]Joint SDU-NTU Centre for Artificial Intelligence Research (C-FAIR),
Shandong University, China
duwei@sdu.edu.cn, bywu@mail.sdu.edu.cn,
{sun_yuqing, guowei, yuntaodu, yzm, gxyu, clz}@sdu.edu.cn

## Abstract

Effective communication is crucial for multi-agent cooperation in partially observable environments. However, a fundamental trilemma exists among task performance, communication efficiency, and human interpretability. To resolve this, we propose a multi-agent communication framework via **G**rounding **L**anguage and **C**ontrastive learning (GLC) to learns efficient and interpretable communication protocols. Specifically, GLC employs an autoencoder to learn discretized and compressed communication symbols, ensuring high communication efficiency. These symbols are then semantically aligned with human concepts using data generated by a Large Language Model (LLM), making them human-interpretable. Furthermore, a contrastive learning objective is introduced to ensure consistency and mutual intelligibility among all agents, thereby securing high task utility. GLC dynamically balances these objectives by the Information Bottleneck principle. Extensive experiments show that GLC outperforms state-of-the-art methods across multiple benchmarks, delivering superior task performance, higher communication efficiency, and enhanced human interpretability.

## 1 Introduction

Multi-Agent Reinforcement Learning (MARL) has achieved remarkable success in various complex real-world multi-agent systems (MAS) such as autonomous driving Xu et al. (2024) and traffic signal control Zhang et al. (2024). In MAS under partial observability, communication learning is key to overcoming individual perceptual limitations and achieving coordinated behavior. However, existing MARL approaches often face a fundamental tension between communication efficiency and human interpretability. Methods that prioritize efficiency typically produce opaque protocols that hinder collaboration with humans Lin et al. (2021); Freed et al. (2020), while those favoring interpretability often incur high communication costs that limit their practical applicability Lowe et al. (2019).

This challenge can be understood through the lens of the Information Bottleneck (IB) principle, which formalizes the trade-off between the complexity of a message and its informativeness about a source variable. As shown in Figure 1(a), in multi-agent settings, this translates to a three-way balance among task performance (utility), communication efficiency (complexity), and human interpretability (informativeness) Tucker et al. (2022a). An ideal protocol must compress high-dimensional observations into efficient messages (minimizing complexity) while preserving enough information (maximizing informativeness), both to achieve task goals (maximizing utility) and to remain interpretable by humans or unfamiliar agents.

To address this trilemma, we propose a multi-agent communication framework via Grounding Language and Contrastive learning (GLC). As shown in Figure 1(b), GLC integrates 3 key components: a discrete autoencoder that minimizes communication complexity through compressed symbolic representations, a communication alignment mechanism that aligns discrete symbols with human

---

*Corresponding author

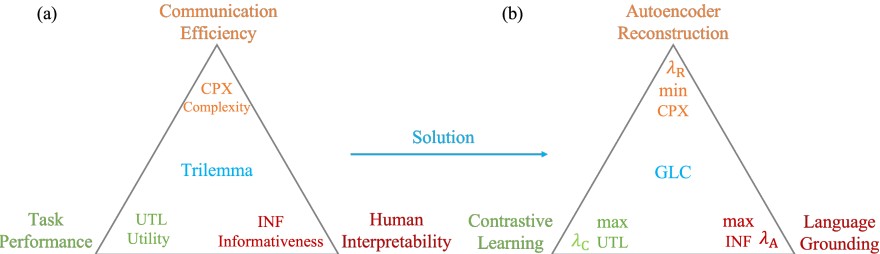

Figure 1: (a) The fundamental trilemma in multi-agent communication, balancing efficiency, performance, and interpretability. (b) The GLC framework addresses this trade-off via autoencoder compression, LLM-based language grounding, and contrastive learning for inter-agent consistency.

semantic spaces to enhance interpretability, and a contrastive learning objective that ensures consistency among agents. By dynamically balancing these objectives, GLC operationalizes the IB principle for MAS to develop communication protocols that are both efficient and interpretable. During execution, agents communicate via discrete symbols, whose rich semantics are derived from LLM supervision and contrastive alignment. A key innovation is the decoupling of training from deployment: while training utilizes LLM-derived anchors and contrastive learning, inference operates entirely without external supervision. Extensive experiments show GLC outperforms baselines in task performance, communication efficiency, and human interpretability. The core contributions are: (1) An integrated framework that learns efficient protocols through discrete compression and grounds them with semantic meaning via language alignment, thereby optimizing the efficiency-utility-interpretability trilemma. (2) A principled training paradigm that unifies contrastive inter-agent alignment with offline LLM grounding, ensuring the emergent language is human-interpretable and consistently shared, leading to more robust and generalizable collaboration.

## 2 RELATED WORK

**Utility-Optimized Communication**. A large body of multi-agent communication work prioritizes maximizing task performance, often at the expense of interpretability and sometimes efficiency. Early methods like CommNet Sukhbaatar et al. (2016) and IC3Net Singh et al. (2019) enabled end-to-end learning of continuous communication vectors that are highly optimized for specific tasks. Subsequent works like TarMAC Das et al. (2019) and MAGIC Niu et al. (2021) introduced attention mechanisms and targeted communication to enhance utility further. While these approaches excel in performance, the learned protocols are typically opaque and require high communication cost, making them unsuitable for human-agent collaboration or bandwidth-constrained environments.

**Efficiency-Driven Communication.** Another line of research focuses on reducing communication cost, often through discretization Carmeli et al. (2023); Foerster et al. (2016). Methods like aeComm Lin et al. (2021) use autoencoders to compress observations into discrete symbols. VQ-VIB Tucker et al. (2022a) further incorporates IB constraints to learn a constrained vocabulary. These methods achieve high communication efficiency but often result in symbols that are not semantically grounded, limiting their interpretability and generalizability to novel partners.

**Interpretability-Oriented Communication** A more recent trend seeks to align agent communication with human-understandable concepts. Some approaches Lazaridou et al. (2020); Tucker et al. (2022b); Li et al. (2024) leverage pre-trained language models or human data to ground continuous communication vectors in natural language semantics. While these methods enhance interpretability, they often rely on continuous representations that incur significant communication overhead, or they lack the structured discrete symbols necessary for efficient communication.

**Bridging the Trilemma.** Prior approaches often excel in only one or two aspects of the trilemma, failing to balance all three. Our GLC framework addresses this gap by integrating discrete symbols for efficiency, language grounding for interpretability, and contrastive learning for utility and consistency into a unified, information-theoretic framework.

## 3 PRELIMINARIES

In this study, we model multi-agent reinforcement learning (MARL) with communication using a decentralized partially observable Markov decision process (Dec-POMDP) framework Oliehoek (2012). The model is formally defined by the tuple $\langle N, \mathcal{S}, \mathcal{O}, \mathcal{C}, \mathcal{A}, \mathcal{T}, R, \gamma \rangle$, where $N$ represents the number of agents; $\mathcal{S}$ is the state space of the environment; $\mathcal{O} = \mathcal{O}^1, \ldots, \mathcal{O}^N$ corresponds to the observation spaces available to each agent; $\mathcal{C} = \mathcal{C}^1, \ldots, \mathcal{C}^N$ defines the sets of communication messages that agents can transmit; $\mathcal{A} = \mathcal{A}^1, \ldots, \mathcal{A}^N$ denotes the sets of executable actions for each agent; $\mathcal{T} : \mathcal{S} \times \mathcal{A}_1 \times \ldots \times \mathcal{A}_N \to \Delta(\mathcal{S})$ defines the transition funcaion; $R : \mathcal{S} \times \mathcal{A}^1 \times \ldots \times \mathcal{A}^N \to \mathbb{R}$ represents the reward funcion.

At each timestep $t$, agent $i$ receives a local observation $o_i^t$ and a set of communication messages $c^{t-1} = \{c_1^{t-1}, \ldots, c_N^{t-1}\}$ sent by all agents at the previous timestep. Using this information, the agent chooses an action $a_i^t \in \mathcal{A}_i$ and generates a new message $c_i^t \in \mathcal{C}_i$ to be broadcast to others. The state transition dynamics are governed by the function $\mathcal{T} : \mathcal{S} \times \mathcal{A}_1 \times \ldots \times \mathcal{A}_N \to \Delta(\mathcal{S})$, which takes the current $s^t$ and joint action $a^t = \{a_1^t, \ldots, a_N^t\}$ as input, and returns a probability distribution over the next state $s^{t+1}$. Here, $\Delta(\mathcal{S})$ denotes the set of all probability distributions over the state space $\mathcal{S}$, reflecting the possible stochastic outcomes of the joint action. After the state transition, each agent $i$ receives an individual reward $r_i^t \in R(s^t, a^t)$ determined by the reward function.

In this study, we address a fully cooperative multi-agent scenario in which all agents collaborate toward a common goal: maximizing the total expected return of all agents. The team's objective is formally defined as:

$$\max_{\pi_i : \mathcal{O} \to \mathcal{A} \times \mathcal{C}} \left[ \sum_{t=1}^{T} \sum_{i=1}^{N} \gamma^t r_i^t \mid (s_i^t, a_i^t) \sim \pi_i, o_i^t \sim \mathcal{O} \right], \tag{1}$$

where $T$ is the timestep horizon and $\gamma$ is the discount factor. To optimize this objective, we adopt policy gradient algorithms, with a particular focus on the asynchronous advantage actor-critic (A3C) framework Mnih et al. (2016). We further enhance the learning stability and credit assignment using Generalized Advantage Estimation (GAE) Schulman et al. (2015), which helps in reducing variance while maintaining tractable bias in advantage estimates.

Regarding the language learning objective for the agents, we follow the conceptualization of language acquisition introduced in Li et al. (2024). Specifically, a target language $L^* \in \mathcal{L}$ is defined as the communication language that optimally enhances collective task performance. Here, $\mathcal{L}$ denotes the space of all possible natural languages. Formally, any language $L \in \mathcal{L}$ can be understood as a set of communication messages $\mathcal{C}$, established through a mapping $L : \Omega \to \mathcal{C}$ that encodes agent observations into messages. In reinforcement learning terms, this is analogous to the process of generating natural language descriptions from input observation vectors. To support language learning, we build a training dataset $\mathcal{D}$ consisting of $|\mathcal{D}|$ (observation, action) pairs. These are extracted from expert trajectories produced by LLM agents communicating via $L^*$. The central aim is to enable trained agents to use $L^*$ effectively for seamless coordination with experts in ad-hoc teamwork settings. A critical requirement is that the acquired language capability exhibits strong generalization, such as robust performance in previously unseen scenarios not present in $\mathcal{D}$.

## 4 METHODOLOGY

This section presents GLC, a MARL framework designed for learning efficient and interpretable multi-agent communication. As illustrated in Figure 2, the framework integrates four core modules that work together to address the trilemma involving performance, efficiency, and interpretability: (1) The MARL agent module, which learns policies and generates efficient communication symbols via a discrete autoencoder; (2) The LLM agent module, which interacts with the environment in text space to produce expert trajectories rich in semantics; (3) The MARL-LLM language grounding module, which aligns the semantics of discrete symbols with natural language descriptions generated by the LLM; (4) The communication alignement contrastive learning module, which ensures all agents develop a consistent and mutually understandable communication protocol. These modules are jointly optimized through a unified multi-objective loss function. Each module will be elaborated

in detail below. The GLC framework is grounded in the IB principle, a detailed derivation of the connection between GLC objectives and the IB principle is provided in the Appendix A.1 and A.2.

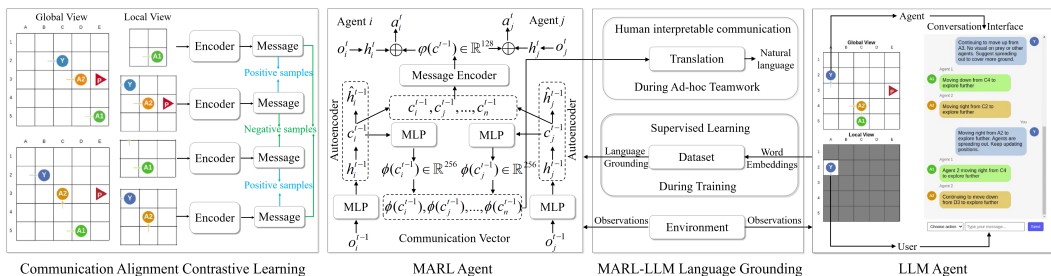

Figure 2: The GLC framework integrates four core modules: (1) MARL Agents: encode partial observations into a discrete symbols via an autoencoder for efficient communication; (2) LLM Agents: interact in a textual space to generate expert trajectories that provide semantically grounded messages Li et al. (2024); (3) MARL-LLM Language Grounding: aligns the discrete embeddings with LLM-generated message embeddings using a cosine similarity loss; (4) Communication Alignment Contrastive Learning: ensures a consistent protocol across all agents by a contrastive loss.

## 4.1 MARL AGENT

At timestep $t-1$, each MARL agent $i$ receives its observation $o_i^{t-1}$ and generates a discrete communication message $c_i^{t-1}$. The observation is first processed through an MLP encoder to obtain a 128-dimensional feature vector $h_i^{t-1} \in \mathbb{R}^{128}$. This feature is then passed through a communication autoencoder to produce discrete symbols $h_i^{t-1} \rightarrow c_i^{t-1}$. The autoencoder consists of an encoder and a decoder, both implemented as a 3-layer multilayer perceptron (MLP). The decoder reconstructs the original feature from the discrete message via mapping $c_i^{t-1} \rightarrow \hat{h}_i^{t-1}$, with quantization applied prior to decoding. To enable gradient backpropagation through the discretization step, a straight-through estimator is employed Bengio et al. (2013). An auxiliary reconstruction loss $\mathcal{L}_{\text{recon}} = ||h_i^{t-1} - \hat{h}_i^{t-1}||_2^2$ is minimized alongside the policy gradient loss $\mathcal{L}_{\text{policy}}$ during training.

At the following timestep $t$, each agent utilizes a shared message encoder to embed the previously transmitted communication symbols $c^{t-1} = c_1^{t-1}, \ldots, c_N^{t-1}$ through linear projection. These message embeddings are concatenated and processed by a 3-layer MLP, resulting in a fixed-dimensional message representation $\varphi(c^{t-1}) \in \mathbb{R}^{128}$. Each agent employs an individual policy head implemented as a gated recurrent unit (GRU) module Chung et al. (2014) with a linear output layer. The policy network combines the agent's encoded state feature $h_i^t$ with the message representation $\varphi(c^{t-1})$ via feature concatenation: $\varphi_i^{t-1} = h_i^t \circ \varphi(c^{t-1})$, where $\circ$ denotes concatenation along the feature dimension. The GRU-based policy subsequently produces both an action probability distribution $a_i^t \sim \pi(\varphi_i^t)$ and corresponding value estimates. These outputs are utilized to compute the policy loss $\mathcal{L}_{\text{policy}}$, which is optimized concurrently with the reconstruction loss $\mathcal{L}_{\text{recon}}$.

## 4.2 LLM AGENT

In our framework, we utilize LLM-based embodied agents to collect samples of the target language $L^*$ in accordance with the LangGround Li et al. (2024). To support environment interaction, we introduce a textual interface $I$, as proposed in Li et al. (2023), that enables two-way conversion between natural language descriptions and structured abstract representations. All $n$ LLM agents operate under generalized task instructions that guide them to collaboratively achieve common goals. At each timestep $t$, agent $i$ receives an English description $I(o_i^t)$ of its current observation, which also includes communicated messages from other agents at the previous timestep $I(C^{t-1})$. Each agent then generates both a communication message and an action decision, which are encoded into abstract representations $C_i^t$ and $A_i^t$ for subsequent task execution. We formalize a dual-environment framework consisting of a textual space and a physical task space, ensuring informational equivalence between LLM and MARL agents despite their representational discrepancies.

Notably, the action-selection and communication behaviors of LLM agents arise intrinsically from their pre-trained capabilities, as we intentionally avoid providing explicit coordination guidance in the prompts. The resulting LLM-generated expert trajectories constitute a supervised dataset $\mathcal{D}$, which captures mappings from individual agents' (observation, action) pairs to natural language messages. Throughout MARL training, $\mathcal{D}$ supplies grounded language signals that steer the emerging communication protocols toward human-interpretable patterns. Further details on LLM agent setup and trajectory collection are provided in the Appendix A.4.

### 4.3 MARL-LLM Language Grounding

To align communication between MARL and LLM agents, each MARL agent first maps discrete communication symbols $c_i^t$ to a continuous vector representation $m_i^t = \phi(c_i^t) \in \mathbb{R}^{256}$ using a 3-layer MLP. To steer the emerging communication protocols toward human-interpretable patterns, we introduce a supervised language alignment loss during MARL training. At each timestep, contextually relevant human-language reference messages $C(o_i^t, a_i^t)$ are retrieved from the dataset based on the agent's current observation and action, providing semantic guidance for communication learning.

To promote semantic consistency between the emergent communication protocol and natural language representations, we design a similarity-driven objective function within a shared embedding space. For each agent's communication vector $m_i = \phi(c_i^t)$, we compute the cosine similarity between $m_i$ and the corresponding human-language reference embedding $m_r$ when supervision is available in dataset $\mathcal{D}$. $m_r$ represents the message embedding the LLM agent produced at that exact timestep in the collected trajectory. This results in the following conditional alignment loss:

$$\mathcal{L}_{align} = \mathbb{I}_{\mathcal{D}}(o_i^t, a_i^t) \cdot \left[ 1 - \frac{(m_i^t)^\top m_r}{\|(m_i^t)\| \cdot \|m_r\|} \right], \tag{2}$$

where the indicator function $\mathbb{I}_{\mathcal{D}}$ enables the loss only for state-action pairs available in the expert dataset, using $C(o_i^t, a_i^t)$ to supply the target embedding $m_r$ in such supervised instances. This explicitly maps both the agent's communication messages and natural language references into a shared high-dimensional semantic space, where geometric consistency is enforced via cosine similarity.

### 4.4 Communication Alignment Contrastive Learning

Drawing on ideas from Lo et al. (2023), this loss encourages semantic consistency among messages generated by different agents observing the same environmental state. For a given message $m_i^t = \phi(c_i^t)$, we define its positive set as messages produced by all other agents within a temporal window $w$ around timestep $t$ in the same trajectory $\tau$:

$$H\left(m_t^i\right) \equiv \left\{ \phi\left(c_j^{t'}\right) \mid j \neq i, t' \in [t-w, t+w] \right\}. \tag{3}$$

Negatives consist of messages sampled from other trajectories within the same training batch. Let $Z(m_i^t) \equiv M \setminus m_i^t$ denote the set of all messages in the batch excluding $m_i^t$, where $M$ comprises all messages across the batch of trajectories. We optimize the following supervised contrastive loss:

$$\mathcal{L}_{contra} = \sum_{m_i^t} \frac{-1}{|H(m_i^t)|} \sum_{m_h \in H(m_i^t)} \log \frac{\exp\left(m_i^t \cdot m_h / \rho\right)}{\sum_{m_z \in Z(m_i^t)} \exp\left(m_i^t \cdot m_z / \rho\right)}, \tag{4}$$

where $\rho \in \mathbb{R}^+$ denotes a temperature parameter that scales the similarity distribution, and $|H(m_i^t)|$ indicates the number of elements in the positive set. During training, each agent maintains a replay buffer that stores trajectory data gathered across multiple environment instances. This buffer includes communication messages received during interaction, which are used to compute the contrastive alignment loss. Based on hyperparameter tuning over different window sizes, we set the temporal context window to 5 timesteps for all environments. Following Khosla et al. (2020), all message embeddings are normalized prior to loss calculation, and a low temperature value ($\rho = 0.1$) is applied to enhance training stability and empirical performance.

## 4.5 DYNAMIC BALANCING OF OBJECTIVES

The communication protocol is optimized via a multi-objective learning framework integrating four complementary goals: (1) policy gradients that learn to share strategically relevant information across agents; (2) linguistic alignment with human-like communication examples drawn from $\mathcal{D}$; (3) autoencoder-based reconstruction of message semantics; and (4) contrastive learning for communication alignment. These components are combined into a unified optimization objective through the following composite loss:

$$\mathcal{L} = \mathcal{L}_{policy} + \lambda_A \mathcal{L}_{align} + \lambda_R \mathcal{L}_{recon} + \lambda_C \mathcal{L}_{contra}. \qquad (5)$$

The weighting coefficients $\lambda_A$, $\lambda_R$, and $\lambda_C$ balance the contributions of the alignment loss, reconstruction loss, and contrastive loss, respectively. Crucially, instead of treating these as static hyperparameters, we view them as dynamic controls that guide the emergence of the communication protocol. Inspired by the IB principle's use of annealing Tishby et al. (2000); Tucker et al. (2022a), we employ a scheduling strategy that initially prioritizes learning a shared, meaningful semantics before gradually increasing pressure for efficient compression. This allows the communication protocol to evolve adaptively. The scheduling strategy of coefficients is set as follows,

(1) Task-Adaptive Alignment $\lambda_A$: The weight $\lambda_A$ is set task-dependently. A higher value prioritizes human interpretability in complex tasks like USAR, while a lower value in tasks like Predator-Prey allows greater focus on task performance. (2) Annealing for Communication Efficiency $\lambda_R$: We apply a linear annealing schedule to $\lambda_R$, increasing it from a small value to a higher one. This implements an "explore then compress" strategy: initially allowing rich semantic exploration before gradually increasing pressure for efficient discrete compression. (3) Stabilizing Consensus $\lambda_C$: The $\lambda_C$ is kept at a fixed, moderate value to provide a consistent signal that fosters a shared protocol among agents, ensuring stability during training. In practice, a lightweight scheduler module updates these weights during training. This dynamic balancing enables GLC to adaptively evolve communication protocols optimized for each task's specific requirements and learning stage, rather than settling for a static compromise.

## 5 EXPERIMENTS

In this section, we conduct a comprehensive evaluation of GLC in two multi-agent benchmarks, Predator-Prey Singh et al. (2019) and USAR Li et al. (2023), comparing it against state-of-the-art baselines to rigorously answer the following questions:

**Q1** (Task performance): How does GLC's performance compared with baseline methods?

**Q2** (Communication Efficiency): How does GLC's efficiency compared with baseline methods?

**Q3** (Interpretability): To what extent are GLC's communication protocols human-interpretable?

**Q4** (Trade-off Adaptation): How dynamically select weightings of objectives in diverse tasks?

**Q5** (Generalization): How well does GLC generalize to unseen agent teams?

**Q6** (Contribution): What are the contributions of GLC's components to overall performance?

**Q7** (Scalability): What is GLC's performance trend as the number of agents increases?

All experiments were executed on an RTX 4090 GPU with 24GB Memory. MARL baseline methods were implemented using their officially released code. Through extensive empirical tuning, all models were trained under a unified set of hyperparameters. Each training epoch used a batch size of 500 with 10 model update iterations. Learning rates were set to 0.001 for Predator-Prey and 0.0001 for USAR. The Predator-Prey vision=1 ($pp_{v1}$) task required 2.5 million timesteps and completed in about 1 hour, while the Predator-Prey vision=0 ($pp_{v0}$) and USAR tasks needed 10 million timesteps, taking approximately 3 hours. Baseline methods include IC3Net Singh et al. (2019), aeComm Lin et al. (2021), LangGround Li et al. (2024), VQ-VIB Tucker et al. (2022a), and a non-communicating independent agent baseline (noComm). **Full environment details, baseline descriptions, and hyperparameter settings are available in the Appendix A.3 and A.4.**

**Task Performance (Q1).** Our evaluation assesses GLC's task performance against state-of-the-art baselines. As shown in Figure 3, GLC demonstrates accelerated convergence and improved stability relative to other methods. In the moderately complex $pp_{v1}$ setting, GLC quickly reaches performance levels comparable to leading baselines (IC3Net, aeComm, LangGround), while also exhibiting smoother learning curves. Its advantage is further emphasized in the more difficult $pp_{v0}$ environment. As shown in the middle figure, limited visibility increases the demand for coordinated communication. Despite this, GLC achieves notable performance gains earlier in the training process, highlighting its ability to capture and convey essential semantic information. Most importantly, in the challenging USAR setting, GLC consistently attains higher performance levels with significantly reduced training fluctuations. This performance advantage stems from GLC's ability to learn semantically grounded and mutually consistent communication, which enables more efficient and robust coordination among agents compared to baselines.

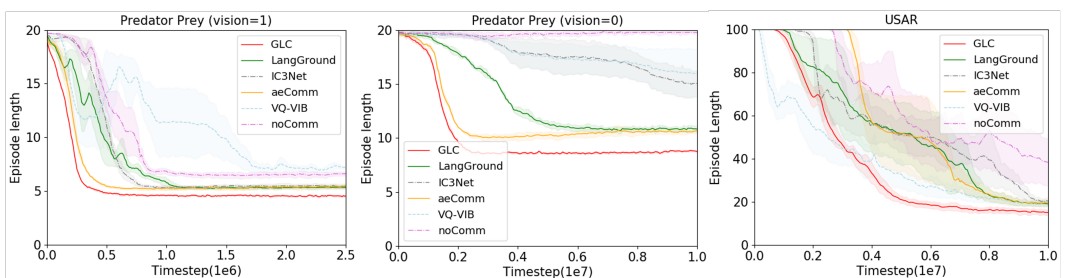

Figure 3: Learning curves of GLC and baselines. The x-axis indicates training timesteps, while the y-axis measures task performance using episode length, with shorter values denoting better results. Shaded regions around each curve reflect standard errors derived from three random seeds.

**Communication Efficiency (Q2).** To evaluate communication efficiency, we measured the total communication cost required to complete tasks across three multi-agent collaboration environments. As summarized in Table 1 (using the $pp_{v1}$ environment as an example), GLC exhibits a clear advantage in communication efficiency. Through discrete symbol compression, GLC reduces the per-step communication volume by several orders of magnitude compared to continuous vector-based methods such as LangGround and IC3Net, achieving exceptionally low communication overhead. This ultra-low bitrate makes GLC particularly suitable for real-world applications with limited bandwidth. Although other discrete approaches, including aeComm and VQ-VIB, also achieve low communication usage per step, GLC maintains stronger semantic expressiveness while retaining a high compression rate, which is essential for communication that is both efficient and interpretable.

GLC also significantly reduces the number of steps required to complete tasks, thereby further lowering the overall communication cost. For instance, in the $pp_{v1}$, GLC agents finish tasks in approximately 4.5 steps on average, while baseline methods require between 5.3 and 7.2 steps. Combined with its low per-step transmission of only 32 bits, GLC's total communication cost is substantially lower than all compared methods. In summary, GLC not only achieves high per-message efficiency through discrete compression but also reduces the need for repeated communication through more effective collaborative behavior. These characteristics make it particularly suitable for deployment in bandwidth-constrained scenarios where both interpretability and operational efficiency are critical. Additional results for the other environments are provided in the Appendix A.3.

Table 1: Maximum Theoretical Communication Bits per Agent per Task Completion in $pp_{v1}$ environment (Assuming no gating optimization)

| Method | Bits/Step | Avg. Steps | Total Bits | Ratio to GLC |
|---|---|---|---|---|
| GLC | 32.0 | 4.5 | 144.0 | 1.0 |
| LangGround | 8192.0 | 5.3 | 43417.6 | 301.5 |
| IC3Net | 8192.0 | 5.5 | 45056.0 | 312.9 |
| aeComm | 24.0 | 5.4 | 129.6 | 0.9 |
| VQ-VIB | 58.0 | 7.2 | 417.6 | 2.9 |
| NoComm | 0.0 | 6.6 | 0.0 | 0.0 |

**Interpretability (Q3).** We next evaluate the semantic structure of the learned communication space, with a focus on its human-interpretable properties and similarity to natural language. To evaluate how effectively GLC aligns agent communication with human language, we conduct a quantitative assessment of the interpretability of its grounded messages. Using the offline dataset $\mathcal{D}$ as reference, we compute two metrics: (1) the embedding-space similarity between GLC agent messages and LLM agent responses in identical $pp_{v0}$ scenarios over 100 evaluation episodes, and (2) the BLEU score between GLC messages and their closest semantic matches in $\mathcal{D}$. As presented in Table 2, GLC slightly surpasses LangGround on both measures. Methods without explicit language alignment exhibit near-random interpretability results and are therefore omitted from comparison.

Table 2: Interpretability Metrics Comparison

|  | Cos sim | | Bleu score | |
| --- | --- | --- | --- | --- |
|  | GLC | LangGround | GLC | LangGround |
| $pp_{v0}$ | **0.87±0.02** | 0.82±0.02 | **0.65±0.04** | 0.52±0.03 |
| $pp_{v1}$ | **0.86±0.03** | 0.81±0.03 | **0.54±0.10** | 0.45±0.12 |
| USAR | **0.84±0.07** | 0.79±0.12 | **0.51±0.05** | 0.42±0.04 |

To assess semantic organization, we apply clustering and visualization to message embeddings collected over 100 evaluation episodes in the $pp_{v0}$ environment, following the procedure in Lin et al. (2021). Using t-SNE Van der Maaten & Hinton (2008) for dimensionality reduction and DBSCAN Ester et al. (1996) for cluster analysis, we analyze whether the emergent protocol displays meaningful semantic structure. Figure 4 illustrates that the emergent messages form semantic clusters closely associated with specific environmental states. For instance, the rightmost red cluster corresponds to agents located at coordinate (B, 3) without visibility of the prey. Querying the dataset $\mathcal{D}$ returns minimal-distance matches such as the natural language description "moving right from (B, 3)", which accurately captures the agent's situational context. This close correspondence between learned communication patterns and human-interpretable semantics confirms the meaningful organization of the emergent protocol space.

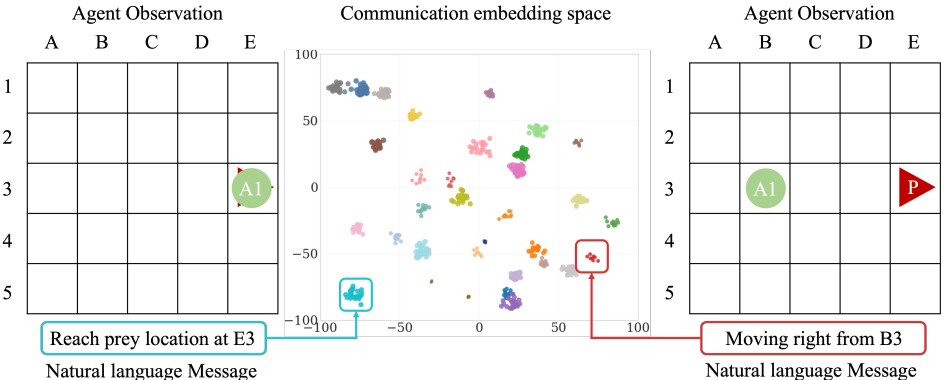

Figure 4: Learned communication embedding space. The communication vectors from $pp_{v0}$ environment are visualized using t-SNE and clustered with DBSCAN. As examples, we identify two semantically meaningful clusters, each corresponding to a specific agent observation. To demonstrate the alignment between the agent communication space and the human language embedding space, we also present the most similar specific reference message from the dataset $\mathcal{D}$.

**Trade-off Adaptation (Q4).** Our weighting strategy is dynamic in two respects: through task-specific presetting of $\lambda_A$ and $\lambda_C$ and training-adaptive scheduling of $\lambda_R$. To validate the efficacy of the latter adaptive schedule, we contrast it with a baseline that uses the same preset weights but keeps them frozen during training. Using the $ppv_0$ task as an instance, the baseline used a fixed set of weights ($\lambda_A = 0.5$, $\lambda_R = 0.1$, $\lambda_C = 0.1$) that remained constant throughout the training. These values were selected based on experimental results in the Appendix A.3. In contrast, the training-adaptive schedule used the linear annealing schedule for $\lambda_R$, starting from $\lambda_R = 0.01$ and annealing to $\lambda_R = 0.1$, while the other two weights consistent with the task-specific presetting weights.

In the $pp_{v0}$ environment, comparative results presented in Table 3 indicate that GLC with dynamic weighting achieves better task performance than the static baseline, completing episodes more efficiently. More notably, it accomplishes this while significantly reducing overall communication cost, demonstrating an ability to learn a more compressed communication protocol without compromising task success. Although there is a slight decrease in interpretability metrics, the protocol remains substantially more understandable than non-grounded baselines, preserving human-interpretable semantics despite higher compression. These findings provide qualitative evidence that the dynamic annealing strategy successfully guides the system through a "learn then compress" trajectory, effectively balancing utility and complexity.

Table 3: Ablation study on dynamic vs. static weighting in the $pp_{v0}$ environment.

| Method | Episode Length ↓ | Total Bits ↓ | BLEU Score ↑ |
|---|---|---|---|
| GLC (Fixed Weights) | $9.62 \pm 0.03$ | $307.8 \pm 0.96$ | $0.65 \pm 0.04$ |
| **GLC (Dynamic Weights)** | $\mathbf{8.71 \pm 0.04}$ | $\mathbf{278.7 \pm 1.28}$ | $0.62 \pm 0.05$ |

Our experimental environments, by design, impose distinct pressures on the communication protocol, emphasizing different corners of the efficiency-utility-interpretability trilemma (as shown in Figure 1). The emergent properties of GLC's protocol naturally adapt to these pressures. In the $pp_{v0}$ environment, where agents are 'blind' and rely solely on communication for survival, the pressure is overwhelmingly toward maximizing task utility. The protocol is driven to convey the most critical information for coordinated search and capture under extreme uncertainty. While efficiency is necessary, the primary imperative is to achieve the task goal, leading to a protocol that is highly effective for coordination even if not maximally compressed. Conversely, the $pp_{v1}$ environment provides partial observability, shifting the primary pressure toward optimizing communication efficiency. With local vision reducing the absolute dependency on communication for basic survival, the value of efficient information sharing becomes paramount. Here, GLC learns a protocol that prioritizes minimalistic, bandwidth-efficient symbols to supplement the agents' own perceptions, achieving the task goal with minimal communicative overhead. Finally, the USAR environment, with its heterogeneous roles and complex action sequences, demands a protocol rich in semantic interpretability. Agents must communicate nuanced intentions, requests, and contextual information (e.g., bomb defusal sequences). Consequently, GLC's protocol in this setting aligns most closely with human language concepts, sacrificing some raw efficiency for the clarity and unambiguous understanding required for successful collaboration in such an intricate task. This demonstrates that GLC does not seek a single optimal point on the trilemma but rather dynamically adapts its emergent communication protocol to the specific constraints and requirements of the task at hand. Further experimental analysis supporting these findings is provided in the Appendix A.3.

## 6 CONCLUSIONS

This paper introduced GLC, a novel multi-agent communication framework that effectively bridges the long-standing trade-off among task performance, communication efficiency, and human interpretability. By integrating discrete autoencoder-based compression with LLM-grounded semantic alignment and inter-agent contrastive learning, GLC enables the emergence of protocols that are simultaneously communication-efficient, task-effective, and human-interpretable. Crucially, a dynamic weighting schedule allows the framework to adaptively balance these objectives throughout training, guided by the specific constraints of the task environment. This adaptability ensures the practical applicability of GLC in diverse real-world domains, including robotic swarms where low-bandwidth communication is paramount, and autonomous vehicle fleets where interpretability is crucial for human trust and collaboration.

While GLC represents a significant step forward, several promising directions remain for future work. These include developing dynamic alignment mechanisms that incorporate real-time human feedback, extending the framework to incorporate multimodal signals, and integrating structured semantic constraints or external knowledge graphs to improve the generalization. A deeper theoretical analysis of the generalization property of emergent communication under grounded learning would also be valuable. Addressing these challenges will further enhance the practicality of GLC for real-world human-AI collaboration and advance the broader goal of developing efficient and interpretable multi-agent systems.

## 7 ETHICS STATEMENT

We promise that we have read the ICLR Code of Ethics, and this article has not raised any questions regarding the Code of Ethics.

## 8 ACKNOWLEDGMENTS

This work was supported by the National Natural Science Foundation of China under Grant (No. 92367202 and 62531013 and 62432006), the Shandong Provincial Natural Science Foundation (No. ZR2024JQ001 and ZR2025LZH004), the Taishan Scholars Program (No. tsqn202408317), Shandong Postdoctora1 Science Foundation (No. SDCX-ZG-202501019), China Postdoctoral Science Foundation (No. 2025M771502)

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

## A APPENDIX

### A.1 INFORMATION-THEORETIC FOUNDATION

The GLC framework is grounded in the IB principle, which formalizes the trade-off between the compression of input data and the preservation of task-relevant information. We frame the learning of communication protocols within this principle: each agent must compress its encoded observation $H_i$ into a concise message $C_i$, while ensuring that the message remains informative about key variables essential for collaboration. Specifically, the message should be informative about (1) Task Utility ($Y$), to enable effective coordination; (2) Human Interpretability ($L$), by aligning with semantic concepts from a language space; and (3) Other Agents' Perspectives ($C_{-i}$), to ensure consistency and mutual intelligibility.

The IB objective of maximizing $I(C_i; Y, L, C_{-i})$ while minimizing $I(C_i; H_i)$ is intractable to optimize directly. Instead, GLC operationalizes this principle through a composite loss function that integrates four complementary objectives, each acting as a surrogate for a component of the IB trade-off:

$$\underbrace{\mathcal{L}_{\text{policy}}}_{-I(C_i;Y)} + \lambda_A \underbrace{\mathcal{L}_{\text{align}}}_{-I(C_i;L)} + \lambda_C \underbrace{\mathcal{L}_{\text{contra}}}_{-I(C_i;C_{-i})} + \lambda_R \underbrace{\mathcal{L}_{\text{recon}}}_{\approx I(H_i;C_i)} \tag{6}$$

Intuitively, the policy loss $\mathcal{L}_{\text{policy}}$ encourages messages to be informative about task success; the alignment loss $\mathcal{L}_{\text{align}}$ grounds them in human-understandable semantics; the contrastive loss $\mathcal{L}_{\text{contra}}$ fosters consensus among agents; and the reconstruction loss $\mathcal{L}_{\text{recon}}$ regulates the compression process. The dynamic balancing of coefficients $\lambda_A, \lambda_R, \lambda_C$ allows GLC to adaptively prioritize different facets of this trade-off during training. A detailed derivation of the connection between the loss function and the IB principle is provided in Appendix A.2.

### A.2 DETAILED INFORMATION-THEORETIC DERIVATION

This section provides a detailed derivation of the connection between the GLC loss function and the IB principle. The core IB objective for an agent $i$ can be formulated as finding a communication message $C_i$ that maximizes the following Lagrangian:

$$\mathcal{L}_{\text{IB}} = I(C_i; Y, L, C_{-i}) - \beta I(C_i; H_i) \tag{7}$$

where $H_i = f_{\text{enc}}(O_i)$ is the encoded representation of observation $O_i$, $I(\cdot; \cdot)$ denotes mutual information, and $\beta$ is a Lagrange multiplier controlling the trade-off.

This objective is intractable for complex environments. We thus decompose it into tractable surrogate losses. First, we note that the information preservation term can be lower-bounded by considering the contributions of each target variable individually:

$$I(C_i; Y, L, C_{-i}) \geq I(C_i; Y) + I(C_i; L) + I(C_i; C_{-i}) \tag{8}$$

This simplification allows us to address each information term separately. The GLC loss function (Equation 6) is designed to maximize these terms:

$$-\mathcal{L}_{\text{policy}} \propto I(C_i; Y) \quad \text{(Task utility)}$$
$$-\lambda_A \mathcal{L}_{\text{align}} \propto I(C_i; L) \quad \text{(Human interpretability)}$$
$$-\lambda_C \mathcal{L}_{\text{contra}} \propto I(C_i; C_{-i}) \quad \text{(Inter-agent consistency)}$$

Conversely, the compression term $I(C_i; H_i)$ is regulated by the reconstruction loss $\mathcal{L}_{\text{recon}}$.

By enforcing the message $C_i$ to be predictive of the By enforcing the message $C_i$ to be predictive of the encoded observation $H_i$ via the autoencoder, we implicitly control the complexity of the message, ensuring that it does not retain excessive, irrelevant information from $H_i$. Since $H_i$ is itself a compressed representation of $O_i$ obtained through the encoder $f_{\text{enc}}$, controlling $I(C_i; H_i)$ effectively constrains $I(C_i; O_i)$ through the data processing inequality. Thus, the overall GLC objective can be viewed as a practical and scalable approximation to the idealized IB principle for multi-agent communication, where we directly control the information flow through the pathway $O_i \to H_i \to C_i$ rather than the direct pathway $O_i \to C_i$. This approach maintains the fundamental IB trade-offs while remaining tractable for complex environments.

### A.3 Additional Experiments

### A.3.1 Baselines

We implemented the MARL baseline methods using their publicly available official code. A short overview of each baseline is provided as follows: aeComm Lin et al. (2021) advances multi-agent discrete communication methods by grounding messages in reconstructed observations, exhibiting better performance than end-to-end and inductive bias methods in decentralized contexts. VQ-VIB Tucker et al. (2022a) is a representative human-interpretable communication paradigm, constructing semantic spaces for discrete tokens that demonstrate effective performance in human-agent collaboration scenarios. IC3Net Singh et al. (2019) is a continuous communication method without language grounding, which employs a gating mechanism to dynamically control inter-agent messaging. LangGround Li et al. (2024) represents the pioneering effort in developing human-interpretable multi-agent communication by aligning continuous vectors with natural language semantics.

### A.3.2 Communication Efficiency (Q2)

To further validate the communication efficiency of GLC, we conducted a qualitative analysis in both the USAR and $pp_{v0}$ environments. In the complex USAR setting, which involves multi-room navigation and specialized coordination requirements, GLC demonstrated significant advantages in low communication overhead. The discrete communication protocol enabled agents to maintain effective coordination while using minimal communication resources, in stark contrast to continuous vector-based approaches, which incurred substantially higher communication costs.

The $pp_{v0}$ presented an even more challenging scenario, where agents operated with zero visual perception and relied exclusively on communication for situational awareness. In this setting, GLC's efficiency advantages became particularly evident. The framework successfully supported complete task coordination through compact discrete symbols, demonstrating that meaningful communication can be achieved without the bandwidth burden associated with continuous vector transmission. Across both environments, GLC maintained a consistent pattern of efficient communication without compromising coordination effectiveness. The discrete symbolic approach proved particularly valuable in scenarios requiring frequent information exchange, where the cumulative bandwidth savings became increasingly significant. Furthermore, the semantic grounding of these discrete symbols ensured that communication remained interpretable despite the high compression ratio. GLC demonstrates robust communication efficiency and coordination across diverse environmental constraints. It performs effectively in perception-limited scenarios and complex multi-step tasks, proving its ability to balance communication cost with coordination effectiveness. This makes GLC highly suitable for real-world applications with limited or costly communication resources.

Table 4: Maximum Theoretical Communication Bits per Agent per Task Completion in $pp_{v0}$ (No Gating Optimization)

| Method | Bits/Step | Avg. Steps | Total Bits | Ratio to GLC |
|---|---|---|---|---|
| GLC | 32.0 | 8.7 | 278.4 | 1.0 |
| LangGround | 8192.0 | 10.8 | 88473.6 | 317.8 |
| IC3Net | 8192.0 | 15.0 | 122880.0 | 441.4 |
| aeComm | 24.0 | 10.6 | 254.4 | 0.9 |
| VQ-VIB | 58.0 | 15.9 | 922.2 | 3.3 |
| NoComm | 0.0 | 19.8 | 0.0 | 0.0 |

### A.3.3 Human Interpretability (Q3)

Topographic similarity quantifies the structural alignment between distances in the observation space, such as physical agent locations, and corresponding distances in the communication space, such as embedded message vectors, as described in Brighton & Kirby (2006). This metric reflects the compositionality and generalizability of the emergent communication protocol, since semantically related observations should ideally yield similar communication signals. Following the method outlined in Lazaridou et al. (2018), we compute this measure using 100 evaluation episodes from the $pp_{v0}$ environment. Our procedure involves computing cosine similarities among all communication

Table 5: Maximum Theoretical Communication Bits per Agent per Task Completion in $USAR$ (No Gating Optimization)

| Method | Bits/Step | Avg. Steps | Total Bits | Ratio to GLC |
|---|---|---|---|---|
| GLC | 32.0 | 14.9 | 476.8 | 1.0 |
| LangGround | 8192.0 | 19.1 | 156467.0 | 328.2 |
| IC3Net | 8192.0 | 20.4 | 167116.8 | 350.5 |
| aeComm | 24.0 | 19.4 | 465.6 | 1.0 |
| VQ-VIB | 58.0 | 18.4 | 1067.2 | 2.2 |
| NoComm | 0.0 | 38.3 | 0.0 | 0.0 |

vector pairs and measuring Euclidean distances among all agent position pairs. The topographic similarity score is then obtained as the negative Spearman correlation coefficient $\rho$ between these two sets of distances. As shown in Table 6, our method achieves the highest topographic similarity, with $\rho = 0.73$, among all baseline approaches, indicating that the resulting communication patterns exhibit structural properties most akin to those of human language.

Table 6: Topographic similarity in $pp_{v0}$

| Methods | Topographic Similarity |
|---|---|
| GLC | **0.73±0.11** |
| LangGround | 0.67±0.07 |
| IC3Net | 0.54±0.14 |
| aeComm | 0.37±0.05 |

### A.3.4 TRADE-OFF ADAPTATION (Q4)

We present the analysis of weight selection in GLC (no annealing schedule). The selection of the language alignment weight $\lambda_A$ is highly dependent on task complexity, as evidenced by the data in Table 7. In the simpler Predator-Prey environments $pp_{v0}$ and $pp_{v1}$, a moderate weight of $\lambda_A = 0.5$ yields the best performance (lowest episode length), indicating that a balanced level of semantic guidance is sufficient for effective coordination. However, in the more complex USAR environment, which requires nuanced communication for specialized roles and multi-step sequences, a higher weight of $\lambda_A = 1.0$ is optimal. This demonstrates that complex collaboration tasks benefit significantly from stronger pressure to align the emergent communication with human-interpretable concepts.

Table 7: Episode length of GLC with different $\lambda_A$ on several scenarios

| Scenarios | $\lambda_A = 0.1$ | $\lambda_A = 0.5$ | $\lambda_A = 1$ |
|---|---|---|---|
| $pp_{v0}$ | 9.94±0.05 | **9.62±0.03** | 10.21±0.07 |
| $pp_{v1}$ | 5.63±0.17 | **5.28±0.13** | 5.42±0.16 |
| USAR | 22.63±1.25 | 21.55±1.06 | **20.85±0.73** |

The reconstruction loss weight $\lambda_R$ plays a critical role in balancing communication efficiency against semantic richness. As analyzed in Table 8, which presents episode length under varying $\lambda_R$ values, we observe a clear trend: an intermediate value of $\lambda_R = 0.1$ yields the optimal task performance across all environments. Excessively low values ($\lambda_R = 0.01$) lead to inadequate compression, resulting in less efficient communication and slightly longer episode completion times. Conversely, overly aggressive compression ($\lambda_R = 0.5$) damages the semantic content of the messages, hindering coordination and degrading performance. This validates our dynamic annealing strategy for $\lambda_R$, which allows the model to initially explore a rich semantic space before gradually applying compression pressure, thereby avoiding the pitfalls of either extreme.

The contrastive learning weight $\lambda_C$, analyzed in Table 9, also shows a clear task-dependent trend. For both Predator-Prey settings, a low weight of $\lambda_C = 0.1$ is sufficient to ensure consistency among agents without introducing disruptive noise. Conversely, in the USAR environment, a higher weight of $\lambda_C = 0.5$ leads to the best performance. This suggests that complex environments with heterogeneous agents require a stronger contrastive signal to foster a robust and mutually intelligible

Table 8: Episode length of GLC with different $\lambda_R$ on several scenarios

| Scenarios | $\lambda_R = 0.1$ | $\lambda_R = 0.5$ | $\lambda_R = 1$ |
|---|---|---|---|
| $pp_{v0}$ | **9.62±0.03** | 10.15±0.09 | 10.47±0.11 |
| $pp_{v1}$ | **5.28±0.13** | 5.81±0.20 | 6.02±0.19 |
| USAR | 23.78±1.32 | **20.85±0.73** | 22.06±0.92 |

communication protocol that can handle intricate coordination demands. In summary, the weight choices are not static but are dynamically adapted based on the specific pressures of the task environment, allowing GLC to effectively balance the trilemma of efficiency, utility, and interpretability.

Table 9: Episode length of GLC with different $\lambda_C$ on several scenarios

| Scenarios | $\lambda_C = 0.1$ | $\lambda_C = 0.5$ | $\lambda_C = 1$ |
|---|---|---|---|
| $pp_{v0}$ | **9.62±0.03** | 9.93±0.07 | 10.06±0.10 |
| $pp_{v1}$ | **5.28±0.13** | 5.43±0.18 | 5.75±0.20 |
| USAR | 22.93±1.19 | **20.85±0.73** | 21.74±1.04 |

A.3.5   GENERALIZATION (Q5).

Our framework is designed to support seamless ad-hoc teamwork Mirsky et al. (2020) among previously unfamiliar agents without pre-coordination. To evaluate this capability, we train GLC agents in a 10×10 Predator-Prey environment under varying levels of language grounding (covering 25%, 50%, 75%, and 100% of environmental states). As shown in Table 10, the benefits of semantic alignment generalize to states without explicit grounding during evaluation, and GLC consistently outperforms LangGround across all grounding levels. Our findings confirm that GLC achieves zero-shot alignment between agent communications and human language embeddings. Importantly, GLC organizes the entire communication space semantically, going beyond mere memorization of observation-message pairs. This semantic structure supports interpretable message generation in novel states, even when training involves only a limited set of grounded examples.

We further assess our agents' performance in ad-hoc teamwork scenarios, which involve cooperating with unfamiliar partners without prior coordination. To simulate human-agent collaboration, we form mixed teams consisting of 2 MARL agents and 1 LLM agent serving as a human proxy. Each configuration is rigorously evaluated over 8 episodes under 3 distinct random seeds. Team performance is measured by the number of steps required to complete the task, with fewer steps indicating higher coordination efficiency. Full results, including means and standard deviations for all conditions, are provided in Table 11. GLC proves uniquely effective for ad hoc collaboration, outperforming all baseline methods when teamed with unfamiliar LLM agents. While homogeneous teams (GLC-GLC or LLM-LLM) achieve the highest performance, GLC's key advantage lies in its ability to bridge the protocol gap through language-aligned communication, ensuring superior coordination in mixed ad-hoc teams.

Table 10: Comparative zero-shot generalization evaluation on $pp_{v1}$ between GLC and LangGround.

| | Cos sim | | Bleu score | |
|---|---|---|---|---|
| | GLC | LangGround | GLC | LangGround |
| 100% | **0.83±0.04** | 0.77±0.04 | **0.70±0.09** | 0.63±0.07 |
| 75% | **0.71±0.10** | 0.66±0.10 | **0.54±0.13** | 0.49±0.14 |
| 50% | **0.43±0.11** | 0.30±0.15 | **0.39±0.18** | 0.30±0.18 |
| 25% | **0.25±0.07** | 0.18±0.06 | **0.30±0.12** | 0.22±0.09 |

Figure 5 (left) shows the success rate of GLC agents under varying degrees of language grounding. The results indicate that increased grounding leads to improved team performance and stronger communication alignment. Our findings confirm the emergence of zero-shot alignment between agent communication signals and human language embeddings. Notably, GLC organizes the entire communication space in a semantically structured manner, going beyond mere memorization of observation-message pairs. This semantic organization supports interpretable message generation

Table 11: Ad-hoc teamwork performance, which is measured by task completion steps.

| Team | $pp_{v1}$ | $pp_{v0}$ | USAR |
|---|---|---|---|
| GLC | 3.56±0.79 | 7.63±3.12 | 18.21±2.64 |
| LLM | 6.79±5.12 | 11.47±5.05 | 15.96±3.28 |
| GLC+LLM | **7.85±5.03** | **14.23±3.81** | **18.70±7.26** |
| LangGround+LLM | 8.51±5.76 | 15.52±4.79 | 23.20±10.61 |
| aeComm+LLM | 10.26±6.37 | 17.67±4.63 | 20.34±9.13 |
| noComm+LLM | 10.72±5.84 | 20.21±0.08 | 31.15±9.73 |

in novel states through the learned topological representation, even when training involves only a limited set of grounded examples.

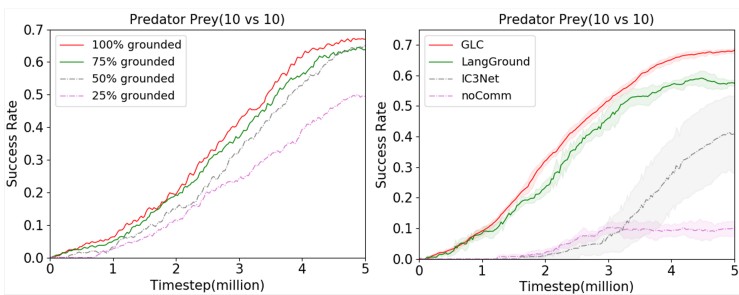

Figure 5: Left: Team performance of GLC agent with different levels of language grounding on $pp_{v1}$ (10 by 10). Right: Comparative performance between GLC and baseline methods in the Predator-Prey (10 by 10) environment with visual range limited to 1.

### A.3.6 CONTRIBUTION (Q6).

We perform an ablation study to assess the architectural contributions of GLC. The framework integrates three core components: an autoencoder for communication compression, an LLM-supervised module for semantic alignment, and a contrastive learning module for enhancing communication consistency and generalization. We evaluate these through three ablated variants: (1) GLC-AE: The autoencoder is removed; discrete symbol generation is replaced with a direct linear projection from observations to continuous vectors; (2) GLC-LLM: The language alignment loss is disabled; while the embedding network is retained, it receives no LLM-supervised training. (3) GLC-CL: The contrastive learning module is ablated.

As shown in Table 12, the ablation study provides several key insights. The performance decline in GLC-AE highlights the essential role of the autoencoder in achieving communication efficiency via learned discrete compression. Although GLC-LLM preserves reasonable task performance, its notably lower BLEU scores emphasize the importance of LLM supervision for generating human-interpretable messages. Additionally, the GLC-CL variant, which removes the contrastive learning component, shows reduced embedding consistency and generalization ability, confirming that contrastive learning is vital for ensuring structural coherence and mutual intelligibility among agents. These results collectively indicate that the autoencoder supports efficient communication, the LLM alignment enables semantic interpretability, and contrastive learning enhances consistency and generalization. The complementary functions of these modules illustrate how GLC effectively balances efficiency and interpretability in multi-agent communication.

Table 12: Ablation study on GLC on $ppv_0$ environment.

| Performance | GLC-CL | GLC-AE | GLC-LLM | GLC |
|---|---|---|---|---|
| Episode length | 11.26±0.07 | 11.02±0.06 | 10.87±0.06 | **9.13±0.05** |
| Cos sim | 0.82±0.03 | 0.84±0.02 | 0.06±0.01 | **0.87±0.02** |
| Bleu score | 0.53±0.06 | 0.56±0.06 | 0.04±0.01 | **0.65±0.04** |

A.3.7 SCALABILITY (Q7).

To assess the scalability of our method, we performed experiments in an extended Predator-Prey setting ($pp_{v1}$, 10×10 grid with 3 predators and 1 prey). Figure 5 (right) presents the learning curves, demonstrating that GLC consistently outperforms all baseline methods. Notably, GLC exceeds the performance of ablative baselines such as IC3Net, which lacks language grounding, and noComm, which uses no communication. To further investigate scalability under more demanding conditions, we extended our analysis to larger grids with proportional increases in agent and prey populations: $pp_{v0}$, 15×15 Grid with 8 Predators and 3 Prey; $pp_{v0}$, 20×20 Grid with 10 Predators and 4 Prey. To ensure computational tractability and define a clear failure state for coordination, a maximum episode length (timeout) was set for each environment: 50 steps for the 15×15 grid, and 60 steps for the 20×20 grid. The comprehensive results, summarized in Tables 13 and 14, underscore the capability of GLC to stabilize emergent communication learning in MARL agents as the scale of the environment increases.

Table 13: Performance Comparison in $pp_{v0}$ (15×15 grids with 8 Predators and 3 Prey), which is measured by task completion steps, success rate and maximum theoretical communication bits per agent per task completion.

| Method | Avg. Episode Length (↓) | Success Rate (↑) | Total Comm Bits (↓) |
|---|---|---|---|
| **GLC** | **25.8** | **79%** | **825.6** |
| LangGround | 35.2 | 58% | 288,358.4 |
| IC3Net | 38.5 | 52% | 315,392.0 |
| aeComm | 29.4 | 68% | **705.6** |
| VQ-VIB | 33.7 | 60% | 1,954.6 |
| noComm | 45.0 | 28% | 0.0 |

Table 14: Performance Comparison in $pp_{v0}$ (20×20 grid with 10 Predators and 4 Prey), which is measured by task completion steps, success rate and maximum theoretical communication bits per agent per task completion.

| Method | Avg. Episode Length (↓) | Success Rate (↑) | Total Comm Bits (↓) |
|---|---|---|---|
| **GLC** | **38.5** | **72%** | **1,232.0** |
| LangGround | 55.2 | 40% | 450,560.0 |
| IC3Net | 57.3 | 35% | 469,401.6 |
| aeComm | 42.6 | 58% | **1,022.4** |
| VQ-VIB | 52.8 | 45% | 3,062.4 |
| noComm | 58.7 | 15% | 0.0 |

As the environment scales in size and population, all methods face increased coordination challenges, leading to a natural decline in task performance. However, GLC demonstrates the most resilient behavior, with the smallest relative degradation in success rate and episode completion efficiency. This robustness stems from its structured communication protocol, which maintains effectiveness even as task complexity grows. In contrast, baseline methods exhibit steeper performance drops. The performance gap between GLC and other methods consistently widens with increasing scale. While all approaches struggle with larger environments, GLC's relative advantage becomes more pronounced in the most challenging settings. This expanding margin demonstrates that the value of semantically grounded communication increases with complexity. The contrastive learning mechanism ensures protocol consistency across large agent populations, while the language grounding provides stable semantic references that enable effective generalization - advantages that become critically important in large-scale coordination scenarios.

Across all tested scales, GLC maintains its exceptional communication efficiency while delivering competitive task performance. The discrete communication protocol avoids the combinatorial explosion that plagues continuous methods, keeping bandwidth requirements manageable even with many agents. This combination of maintained performance and practical efficiency makes GLC particularly suitable for real-world applications where both coordination effectiveness and resource constraints must be considered simultaneously. The framework thus provides a scalable solution that balances the trilemma objectives effectively across different operational scales.

## A.4 Experiments details

### A.4.1 Environment

In the Predator Prey environment, $n$ predators with a restricted visual range $v$ collaborate to locate stationary prey within an $x \times x$ grid. Agents receive a shared reward when any predator reaches the prey, and each episode terminates either when all predators have succeeded or after a maximum of $T$ timesteps. Each predator perceives only a local $v \times v$ grid region and selects movement actions based on these partial observations, making communication essential for effective coordination. Our experiments use a $5 \times 5$ grid with 3 predators and 1 prey under two vision settings: $v = 0$ and $v = 1$. Under $v = 0$, predators perceive the prey only when co-located in the same cell. Each episode is capped at 20 steps. With randomized initial positions and higher-dimensional state and action spaces than standard benchmarks, the Predator-Prey environment presents substantial coordination and perceptual challenges.

The USAR environment Li et al. (2023) simulates a cooperative bomb disposal task in which three specialized agents (Alpha, Bravo, and Charlie) must locate and defuse hidden bombs with unknown color-coded activation sequences. The agents operate in a graph-based environment consisting of $n$ interconnected rooms. Each agent carries unique wire cutters and can execute three types of actions: moving between rooms, inspecting a bomb, or using a cutter. Agents have partial observability, perceiving only the contents of their current room. The action space is combinatorial in nature, scaling with the number of rooms ($n$), available cutters ($m$), and the inspection action. Defusing a bomb with $x$ phases yields a reward of $10 \times x$ points. Episodes end when all bombs are successfully defused or after a timeout. In our implementation, the environment includes $n = 5$ rooms and 5 bombs with varying difficulty: two 1-phase bombs, two 2-phase bombs, and one 3-phase bomb, each assigned one of three possible colors. Each agent is equipped with two distinct wire cutters. The episode terminates after 100 steps if not completed earlier. The task demands precise coordination, as agents must communicate effectively to share essential information such as bomb sequences and cutter availability.

### A.4.2 Embodied LLM agents

We utilize large language models as embodied agents for cooperative team tasks, implementing an architecture that incorporates belief states and communication memory. Each agent maintains an internal record of environmental observations and messages from teammates. The agents operate under minimal task guidelines that deliberately avoid explicit coordination strategies, thereby reducing reliance on extensive prompt engineering and enhancing general applicability. For our experimental setup, we adopt the methodology established in LangGround Li et al. (2024), with further implementation details available in their released resources. We use OpenAI's GPT-4-0125-preview model (temperature=0) via API calls to ensure deterministic and reproducible agent behavior.

We constructed the dataset $\mathcal{D}$ by collecting expert trajectories from GPT-4-based embodied LLM agents during interactive task execution. As shown in Table 11, LLM-only teams achieve performance competitive with MARL methods, confirming that their action-communication policies provide effective guidance for training MARL agents. In the USAR environment, we collected 30 episodes comprising 1500 (observation, action) pairs along with associated communication messages. For the Predator-Prey environments, we gathered 1362 and 1874 pairs for the $pp_{v0}$ and $pp_{v1}$ settings, respectively. To align natural language messages with agent communication vectors, we used OpenAI's text-embedding-3-large API to embed all textual messages into 256-dimensional vectors, matching the dimensionality of the agent communication space.

### A.4.3 LLM-Environment Text Interface

We implement both the Predator-Prey and USAR environments using the Gym API Brockman et al. (2016). To facilitate LLM agent interaction, we develop a textual interface that provides sequential natural language observations and executes corresponding actions, supports natural language message broadcasting with messages appended to subsequent observations, and ensures observational equivalence with MARL agents such that both receive identical partial observations at every timestep. The interface serves as a bidirectional mediator between the game engine and language model agents, translating states into language and language into actions. It constructs natural lan-

guage observations by formatting key game elements, such as round number, team score, visible objects, and received messages, into coherent textual descriptions. For action processing, the system employs keyword-based parsing of agent responses and generates specific error feedback when invalid actions are detected, for example, when an agent attempts to inspect a bomb that does not exist. This dual translation mechanism ensures adherence to environment rules while enabling fluent natural language communication.

### A.4.4 LLM-MARL Collaboration

To simulate human-agent collaboration under constrained resources, we form mixed teams consisting of 2 MARL agents and 1 LLM agent powered by GPT-4-turbo in both Predator-Prey and USAR environments. The LLM agent processes textual observations to produce both actions and communication messages. A dedicated interface converts environment states into natural language descriptions and parses LLM responses into executable actions. Bidirectional communication is facilitated through two mechanisms: messages from the LLM agent are embedded into continuous vectors using OpenAI's API for interpretation by MARL agents, while discrete outputs from MARL agents are translated into natural language via cosine similarity matching against a predefined phrase dataset $\mathcal{D}$. This design enables seamless coordination between learning-based and language-guided agents.

## B DISCUSSION ON SCALABILITY AND FUTURE WORK

Our selection of the Predator-Prey and USAR environments was strategic, as they serve as established and computationally tractable testbeds that effectively capture the core challenges of the efficiency-utility-interpretability trilemma under study. These environments allowed for the extensive ablation studies and convergence analyses necessary to validate GLC's core contributions within practical resource constraints. We acknowledge that evaluation on larger-scale benchmarks like the StarCraft Multi-Agent Challenge (SMAC) or real-world robotic simulators represents a valuable direction for future work, and we confirm that the GLC framework is environment-agnostic and readily generalizable to such scenarios.

The GLC architecture is inherently designed for scalability through several core principles. The discrete autoencoder ensures bandwidth-efficient communication that is invariant to environment size or agent population. Furthermore, the contrastive learning objective maintains semantic consistency and protocol coherence across large agent populations by structuring the communication space based on functional context. The dynamic balancing mechanism, guided by the Information Bottleneck principle, allows the system to adaptively prioritize different objectives—such as compression or semantic richness—depending on the task's complexity and scale. Our scalability experiments in Appendix A.3.7, conducted on enlarged grid worlds with increased agent and prey populations, empirically validate that GLC maintains robust performance and communication efficiency as the problem scale expands. To further demonstrate GLC's generalization ability, we plan to test it in more complex embodied settings such as ALFWorld (multi-step reasoning with natural language) and RoCoBench (grounded multi-agent collaboration). Success in these domains would strongly validate GLC's practicality for real-world human-AI collaboration under longer horizons and physical constraints.

In terms of computational viability, we emphasize that GLC's design is highly efficient and practical for real-world deployment. The use of the LLM is strictly confined to a one-time, offline phase for generating a static dataset of expert trajectories. During the central training and deployment phases, no LLM queries are made, eliminating any ongoing computational overhead, latency, or cost associated with large model inference. This makes GLC particularly suitable for bandwidth-constrained applications like robotic swarms or autonomous vehicle networks, where both interpretability and low communication latency are critical.

GLC creates a synergistic relationship with LLMs rather than seeking to replace them. While LLMs serve as general-purpose knowledge bases and a source of human-aligned semantic grounding, GLC learns task-specific, highly efficient communication protocols. Our ad-hoc teamwork experiments demonstrate that these two paradigms can interoperate effectively, with GLC agents successfully collaborating in mixed teams with LLM agents. This shows that GLC's protocols are not only effi-

cient but also semantically accessible to external human-like intelligences, bridging the gap between opaque RL protocols and verbose natural language.

Looking ahead, our future work will explicitly explore GLC's application in more complex and demanding domains. This includes application to extended multi-agent benchmarks like SMAC, investigation into distributed training strategies to handle increased environmental complexity, and deeper analysis of how the emergent communication vocabulary and its syntactic structure evolve with task difficulty. We are confident that the GLC framework provides a solid and scalable foundation for these future research directions toward practical and interpretable multi-agent systems.

## C  THE USE OF LARGE LANGUAGE MODELS

Large language models (LLMs) were not used for research ideation or writing in this work. An LLM (OpenAI's GPT-4) was employed exclusively for generating the expert trajectory dataset $\mathcal{D}$ used in experiments. The authors take full responsibility for the content of this paper.

