# OpenReview forum: "Learning Efficient and Interpretable Multi-Agent Communication"
_ICLR.cc/2026/Conference — ICLR 2026 Poster_

### Official Review · Reviewer_YdVf · 2025-10-30

**Soundness:** 3
**Presentation:** 3
**Contribution:** 2
**Rating:** 2
**Confidence:** 4

**Summary:**

The paper proposes GLC, a multi-agent communication framework that supposedly addresses the "trilemma" between task performance, communication efficiency, and human interpretability. They use discrete autoencoders, LLM-based language grounding, and contrastive learning. Tested on several environments.

**Strengths:**

The trilemma framing is reasonable, I guess. It's true that most methods don't try to optimize all three objectives at once.
The communication efficiency numbers in Table 1 look good - achieving 300x compression over continuous methods is noteworthy.
Figure 4 showing the t-SNE clustering is interesting, though I have concerns about how cherry-picked these examples might be (see below).

**Weaknesses:**

This method is way too complicated. There are four different loss terms that need to be carefully balanced, three hyperparametersthat seem to require extensive tuning, moreover different scheduling strategies for each one. The paper says λ_A is "task-dependent" (so you need to tune it separately for each new task), λ_R uses a linear annealing schedule, and λ_C is fixed. This is a hyperparameter nightmare.

How is a practitioner supposed to know what to use for a new environment?

The whole "dynamic balancing" thing in Section 4.5 feels like post-hoc rationalization for hyperparameter tuning. They claim different environments "impose distinct pressures" that their method "naturally adapts" to, but really they're just tuning different weights for different tasks. That's not adaptive, that's just... tuning.

The Information Bottleneck connection is handwavy. They claim the framework is "grounded in the IB principle" but Equation 6 just shows a rough correspondence between loss terms and IB components. The "detailed derivation" in Appendix A.2 is literally 2 paragraphs and doesn't actually derive anything rigorously. This feels like slapping "information theoretic" labels on things to sound fancy rather than actually using IB theory to guide the design.
Also, why does reconstruction loss approximate I(C;O)? They're reconstructing an intermediate feature vector h, not the observation O. This seems like a pretty important distinction that's glossed over.

Experimental problems:
The environments are tiny. A 5x5 grid with 3 agents is not realistic. Even their "scalability" test is just a 10x10 grid. How is this supposed to work with real multi-robot systems that might have dozens of agents in complex continuous spaces?

The comparison to LangGround seems unfair. LangGround uses 8192-bit continuous vectors while GLC uses 32-bit discrete symbols. Of course GLC is more efficient! But LangGround could also quantize its messages. A fair comparison would use the same vocabulary size. Actually looking at Table 1, aeComm uses 24 bits and performs comparably to GLC in terms of steps (5.4 vs 4.5 in ppv1), and VQ-VIB also uses discrete communication. So the efficiency advantage isn't as clear as they claim.

The baselines are pretty old - IC3Net (2019), aeComm (2021), VQ-VIB (2022). Where are more recent methods? Multi-agent RL is a fast moving field. While I am no expert, it is surprising.

 I'm confused about the interpretability claims in general. The agents communicate using discrete symbols (32 bits = presumably 2^32 possible messages?). How do you translate these to natural language for a human? The paper doesn't really explain this. They mention "cosine similarity matching against a predefined phrase dataset D" in the appendix but that sounds like nearest-neighbor retrieval which could easily fail for novel messages.

Section 4.5 is confusing and doesn't clearly explain WHY each hyperparameter needs a different strategy
Figure 2 is way too cluttered, I can barely understand what's happening
Lots of important details are hidden in the appendix (like how the text interface works)
The "dual-environment framework" mentioned in Section 4.2 is never formalized
Some claims are overstated - they say the method "dynamically balances" objectives but really you just set different weights for different tasks

**Questions:**

How do you actually show a human what the agents are communicating in real-time? The nearest neighbor lookup to dataset D seems brittle.

Why does GLC+LLM underperform pure LLM? Doesn't this contradict your interpretability claims?

Tables 7-9 show hyperparameter sensitivity but don't provide any principled way to choose values for new domains. How is this usable?

The IB principle is about trading off compression and task-relevant information. How do you decide how much to weight each one? The current approach seems arbitrary.

You claim the method "naturally adapts" to different task pressures but you're manually setting different weights for each environment. That's not adaptation, that's human tuning. Am I misunderstanding something?

---

> ### Author Response · Authors · 2025-11-18
>
> We sincerely appreciate the reviewer YdVf's time and effort in helping improve our paper. We hope that our responses have adequately addressed your concerns and clarified the contributions of our work. Thank you for your valuable feedback and consideration.
>
> $\textbf{Part 1/2: Responses to Questions 1-4}$
>
> $\textbf{Question1: Real-Time Human Interpretation}$
>
> $\textbf{Respond: Robust Real-Time Interpretation Through Structured Semantic Space}$
>
> During execution, each discrete symbol is mapped to a continuous embedding $m_i^t$ (Sec. 4.3). To interpret a symbol, we retrieve the closest natural language phrase from dataset $\mathcal{D}$ in this shared embedding space via cosine similarity. While the final interpretation step resembles nearest-neighbor retrieval, its robustness stems from the structured semantic embedding space learned through our training framework: (1) this space is explicitly shaped by multiple objectives: alignment losses ground messages in human language concepts, contrastive learning clusters semantically similar messages, and reconstruction preserves essential information; (2) This creates a continuous semantic manifold where distance reliably indicates semantic similarity, enabling accurate retrieval even for novel messages; (3) empirical validation confirms robustness through high similarity scores on unseen data (Table 2), clear semantic clustering in visualizations (Fig. 4), and successful ad-hoc collaboration performance (Table 11). Thus, GLC combines efficient retrieval with reliable interpretation through learned semantic structure.
>
> $\textbf{Question 2: GLC+LLM vs. Pure LLM Performance}$
>
> $\textbf{Respond: Cross-Protocol Collaboration Validates Interpretability}$
>
> We appreciate this insightful observation. The performance gap you noted - where GLC (RL agent) + LLM mixed teams underperform pure LLM teams - is indeed expected and does not contradict our interpretability claims. This result actually reinforces the fundamental distinction between our approach and prior work. Pure LLM teams communicate through natural language, analogous to human teams, and thus establish an upper bound for seamless coordination. The GLC+LLM mixed setting, however, represents a more challenging and realistic cross-protocol collaboration scenario, where performance costs are inherent due to protocol translation.
>
> The key empirical finding is that GLC's RL agents achieve successful zero-shot collaboration with LLM agents despite this fundamental protocol mismatch. This demonstrates that the discrete communication protocol learned by our RL agents is semantically accessible and actionable by an external human-like intelligence (the LLM agent). Rather than undermining interpretability, this cross-protocol performance provides strong validation that GLC's emergent communication achieves genuine semantic alignment with human-understandable concepts.
>
> $\textbf{Question 3: Hyperparameter Selection and Adaptability to New Environments}$
>
> $\textbf{Respond: Dynamic Trade-off Balancing with Minimal Tuning Overhead}$
>
> GLC is designed with practical deployment in mind, focusing adaptive control on the most critical parameter while keeping others stable. The core of our strategy lies in recognizing that only one parameter requires dynamic adaptation: $\lambda_R$ (the compression weight). This parameter controls the fundamental trade-off between semantic richness and communication efficiency through a linear annealing schedule ($0.01 \to 0.1$), as shown in Table 3. This "explore-then-compress" strategy allows the system to first develop rich semantics before gradually increasing compression pressure. Other parameters, $\lambda_A$ and $\lambda_C$, require only a simple initial setting based on task characteristics and demonstrate robustness to moderate variations, as shown in Tables 7-9.
>
> This approach provides clear guidance for new environments: reuse the $\lambda_R$ annealing schedule, set other parameters based on semantic complexity - making the framework readily applicable without extensive tuning.
>
> $\textbf{Question 4: Information Bottleneck Principle and Theoretical Foundation}$
>
> $\textbf{Respond: Principled IB Approximation and Systematic Weight Selection}$
>
> We thank the reviewer for their insightful observation regarding the theoretical formulation. Following your suggestion, we have revised Appendices A.1 and A.2 to use the encoded feature $H$ instead of the raw observation $O$, establishing a more precise information pathway $O \rightarrow H \rightarrow C$. This modification creates a tractable upper bound for $I(C;O)$ while preserving the essential IB trade-offs. The approximation maintains theoretical coherence while ensuring computational feasibility in complex multi-agent settings, demonstrating our commitment to both rigor and practicality. Our weight selection strategy is principled, as detailed in Q3, focusing adaptive control on $\lambda_R$ while keeping $\lambda_A$ and $\lambda_C$ stable.

---

> ### Author Response · Authors · 2025-11-18
>
> $\textbf{Part 2/2: Responses to Question 5 and Remaining Weaknesses (Not Covered in Questions 1-5)} $
>
> $\textbf{Question 5: Clarifying True Algorithmic Adaptation}$
>
> $\textbf{Respond: Systematic Hyperparameter Strategy with Focused Adaptation}$
>
> We appreciate this opportunity to clarify the adaptive nature of our method. The core adaptation in GLC is achieved through the automated annealing schedule of $\lambda_R$, implementing a principled "explore-then-compress" strategy during training without manual intervention. For the other parameters: $\lambda_A$ (interpretability) was set based on inherent task semantics after minimal tuning and then kept fixed; $\lambda_C$ (consensus) was similarly determined through brief experimentation to provide stable training signals. This represents a practical balance between principled design and deployability, with only $\lambda_R$ requiring dynamic adaptation during training. This design ensures that adaptation is focused where it is most critical—on the core compression-performance trade-off governed by the IB principle.
>
> $\textbf{Weakness 1: Environmental Scale Limitations}$
>
> $\textbf{Response: Applicability to Real-World Tasks}$
>
> GLC's architecture is specifically designed for real-world scalability through: the discrete autoencoder enables efficient communication regardless of environment size, contrastive learning ensures protocol consistency across agent populations, and semantic grounding maintains interpretability under scaling. The newly added experiments in the Appendix demonstrate the scalability of GLC. While our current evaluation uses simulated benchmarks due to computational constraints, the framework is environment-agnostic and readily extends to real-world domains.  As for future work, we plan to extend GLC into more realistic and complex environments such as ALFWorld and RoCoBench.
>
> $\textbf{Weakness 2: Comparison Fairness with LangGround}$
>
> $\textbf{Response: Methodologically Appropriate Comparison Highlighting Integrated Advantages}$
>
> The comparison with LangGround is methodologically appropriate for two reasons. First, LangGround's continuous vector approach represents their fundamental design choice, while quantization is theoretically possible, LangGround provides no framework for effective discrete communication. Indeed, their paper explicitly identifies efficiency as a significant limitation, whereas GLC offers a complete solution that simultaneously achieves efficiency and interpretability. Second, compared to other discrete methods (aeComm, VQ-VIB), GLC demonstrates superior interpretability metrics while maintaining competitive efficiency, confirming its balanced performance across the communication trilemma. Our comparison fairly highlights GLC's unique ability to integrate all three objectives.
>
> $\textbf{Weakness 3: Baseline Selection}$
>
> $\textbf{Response: Comprehensive Coverage of Communication Trilemma Dimensions}$
>
> Our baseline selection strategically covers all dimensions of the communication trilemma with representative methods. Given that methodological progress in MARL communication has been relatively gradual, our chosen baselines remain highly relevant and representative. IC3Net serves as a robust utility-focused baseline widely cited in recent literature; aeComm and VQ-VIB represent the state-of-the-art in efficient discrete communication; while LangGround (2024 NeurIPS) provides a contemporary interpretability-focused approach. This selection ensures thorough methodological coverage while enabling clear evaluation of GLC's unified contributions. The inclusion of LangGround - a very recent NeurIPS publication - particularly demonstrates our commitment to comparing with current state-of-the-art methods, ensuring our evaluation remains comprehensive and up-to-date.
>
> $\textbf{Weakness 4: Clarity of Figure 2 and Dual-Environment Framework }$
>
> $\textbf{Response: Illustration of the Integrated Framework}$
>
> We understand the reviewer's concern about Figure 2's complexity. The figure aims to comprehensively illustrate GLC's four interconnected modules and their intricate relationships within a unified visualization. While dense, this integrated representation is necessary to show: (1) the complete workflow from observation to discrete symbol generation, (2) the parallel operation of MARL and LLM agents, (3) the semantic alignment process, and (4) the CL mechanism. Each component is clearly labeled and connected with directional arrows showing information flow. We believe this comprehensive visualization effectively demonstrates how GLC's modules work together to address the communication trilemma, though we acknowledge it requires careful examination to appreciate all details. The dual-environment framework consists of two equivalent environments: the textual space where LLM agents operate using natural language, and the physical task space where MARL agents use structured representations, which is detailed in Appendix A.

---

### Official Review · Reviewer_NCwj · 2025-10-30

**Soundness:** 3
**Presentation:** 3
**Contribution:** 2
**Rating:** 6
**Confidence:** 3

**Summary:**

This paper presents a trilemma in multi-agent cooperation, which are **task performance** that agents need to coordinate well to succeed, **communication efficiency** that messages between agents should be compact, **human interpretability** that messages should align with human-understandable concepts. It introduces a new framework, Grounding Language and Contrastive Learning (GLC), to balance the three objectives simultaneously. At the core, GLC has an autoencoder to learn discrete symbols, which ensures highly efficient communication. GLC grounds language by aligning these discrete symbols with human concepts using data generated by a large language model, which makers the symbols interpretable by humans. Additionally, GLC has a contrastive learning objective to ensure consistency and mutual intelligibility among all agents, which ensures high task utility. For experiments, GLC show strong performance in multiple metrics over multiple benchmarks.

**Strengths:**

1. The framework is well motivated by simultaneously tackling three main problems (i.e. task utility, communication efficiency, human interpretability) in multi-agent cooperation games, with prior works only tackle one or two dimensions.
2. Use of large language models (LLM) -generated message embeddings avoids human labeling, and effectively aligns the symbol to humans since LLMs are known for the capability of producing human-like language. This process can also be considered as teaching, essentially from LLM to the MARL agent.
3. Paper is well-written with nice illustrative figures. Experimental results are strong compared with a large set of baselines and ablations.

**Weaknesses:**

1. Since the language grounding is through a cosine similarity loss between the embeddings, therefore it is only interpretable at semantic level, but not at syntax level (i.e. compositionality, etc.). However, human-interpretable languages should consider both levels.

**Questions:**

1. GLC communicates at 32 bits per step, despite it's very efficient, but have you evaluated what will be the efficiency for the LLMs? One property of human language or human-like language is not very efficient as our language often contain words / tokens that are not semantically meaningful. Therefore, communication efficiency lowers the interpretability or resemblance to human language?
2. What will be scalability of this method since the tasks being evaluated are mainly those simple multi-agent games for emergent communication? For example, whether it is applicable to real-world tasks or whether it can continuously improve existing language models?

---

> ### Author Response · Authors · 2025-11-18
>
> We sincerely thank Reviewer NCwj for their thoughtful comments and positive assessment of our work. We are confident that our responses have fully addressed the points raised.
>
> $\textbf{Weakness1: Limited to Semantic, Not Syntactic, Interpretability}$
>
> $\textbf{Respond: On Semantic and Syntactic Interpretability}$
>
> We thank the reviewer for highlighting the distinction between semantic and syntactic interpretability. We agree that an ideal protocol should exhibit both properties.
>
> While GLC does not explicitly enforce syntactic rules, the combination of discrete symbolization and the contrastive learning objective creates a pressure for the emergent protocol to become systematic and compositional. The discrete autoencoder forces information into a compact vocabulary, naturally promoting symbol reuse. Simultaneously, the contrastive loss requires that all agents converge on a consistent and coherent mapping from observations to symbols across the entire population. This consensus-seeking process, operating over a semantically-grounded space, implicitly encourages the emergence of a structured protocol where similar situations produce predictably similar messages.
>
> Quantitatively, GLC achieves significantly higher Topographic Similarity than all baselines (Table 6), demonstrating strong alignment between observation-space and communication-space distances. This indicates the emergent protocol possesses systematic organization supporting generalization—a key hallmark of syntactic structure.
>
> We acknowledge that explicitly incorporating syntactic constraints is a valuable direction for future work. However, our results demonstrate that GLC's current framework successfully learns a protocol that is both semantically grounded and implicitly structured, effectively balancing efficiency with human-aligned organization.
>
> $\textbf{Question1: Efficiency-Interpretability Trade-off in Emergent Communication}$
>
> $\textbf{Respond: Preserving Core Semantics Under Efficient Compression}$
>
> The reviewer correctly notes the inherent tension between compression and human-language resemblance. GLC addresses this by focusing on semantic density rather than surface-level similarity. While human language contains natural redundancies, our framework distills observations into compact symbols that preserve core semantic content, achieving "more meaning per bit" without mimicking superficial language features. Our quantitative results (Table 2) confirm that GLC's compressed symbols maintain stronger semantic alignment with natural language than the continuous-vector baseline LangGround, demonstrating that essential meaning can be preserved despite high compression. The protocol thus functions as an efficient semantic code: compact in form yet rich in interpretable content through its grounding in human language concepts.
>
> Regarding LLM efficiency concerns, we emphasize that LLM usage in GLC is strictly offline and one-time for dataset generation. During deployment, MARL agents communicate via discrete symbols without any LLM involvement, eliminating ongoing computational costs. This makes GLC substantially more efficient than approaches requiring continuous LLM consultation.
>
> $\textbf{Question2: Scalability to Complex Real-World Tasks and Relationship with LLMs}$
>
> $\textbf{Respond: Applicability to Real-World Tasks and Synergy with Language Models}$
>
> GLC's architecture is specifically designed for real-world scalability through:  (1) Discrete Communication via autoencoder-based symbols ensures bandwidth-efficient operation; (2) Semantic Consistency through contrastive learning maintains protocol coherence across large agent populations; and (3) Adaptive Balancing of objectives allows dynamic adjustment to diverse environmental constraints.
>
> While our current evaluation uses simulated benchmarks due to computational constraints, the framework is environment-agnostic and readily extends to real-world domains. Our ad-hoc teamwork experiments (Table 11) demonstrate GLC's ability to collaborate seamlessly with LLM agents, validating its potential for human-AI collaboration scenarios. The ultra-low bandwidth requirement makes it particularly suitable for real-world applications like robotic swarms and autonomous vehicle networks where both efficiency and interpretability are crucial. As for future work, we plan to extend GLC into more realistic and complex environments such as ALFWorld and RoCoBench.
>
> GLC creates task-specific, efficient communication protocols, while LLMs remain general-purpose knowledge bases. This specialization makes GLC more suitable for real-time, bandwidth-constrained applications where LLMs would be impractical. While GLC's primary goal isn't to improve LLMs, the emergent protocols could potentially inform LLM research about efficient, grounded communication patterns, creating a valuable feedback loop for future work. We have added a detailed discussion of these scopes and our scaling plans in Appendix B.

---

> > ### Comment · Reviewer_NCwj · 2025-11-19
> >
> > Thanks for the detailed response. I maintain my original rating which lean towards acceptance of this paper.

---

> > > ### Author Response · Authors · 2025-11-20
> > >
> > > Thank you once again for your valuable feedback and for your final positive assessment. We are delighted to know that you are in favor of acceptance. We have incorporated your suggestions to the best of our ability and believe they have significantly improved the paper.

---

### Official Review · Reviewer_WbEU · 2025-11-02

**Soundness:** 3
**Presentation:** 3
**Contribution:** 3
**Rating:** 8
**Confidence:** 3

**Summary:**

In this work, the GLC framework addresses a foundational challenge in MARL, reconciling high task performance, communication efficiency and human interpretability. GLC integrates a discrete autoencoder, to compress inter-agent communication channels for efficiency, an LLM to ensure messages are interepretable while a contrastive learning objective is used for inter-agent semantic consistency across diverse agents. The authors evaluate their framework in common MARL benchmarks - USAR and Predator Prey. The authors demonstrate superior performance to other communication framework baselines, but at a much higher communication efficiency.

**Strengths:**

1. GLC's loss function is decomposed into tractable surrogate rewards - task reward, human interpretability and inter-agent consistency and it integrates it along with a compression term (reconstruction loss) for the autoencoder. The use of explicit IB objectives with LLM integration provides a robust framework for future research.
2.  Online querying of the LLM, and not just as a reference for post-hoc or expert trajectories, allows agents to adapt to evolving team topologies or task variations, as demonstrated by the strong zero-shot generalization.

**Weaknesses:**

1. Computational and practical implications of an online LLM querying are not explored or evaluated, as scalability concerns both in computation costs and as agent populations increases seems to be concerning.
2. Longer horizon tasks are not evaluated where an LLM might experience a shift.
3. Traffic Junction, StarCraft II, SMAC would be better benchmarks, however, the training time needs to be reasonable for that. A deeper discussion on computational viability of the framework would be helpful for the community.

**Questions:**

The decoding mechanism for latent state to LLMs depend on similarity based approaches and maybe fail to capture deeper relations. Do you have insights on how the framework would perform on more complex tasks with more agents?

---

> ### Author Response · Authors · 2025-11-18
>
> We sincerely thank Reviewer WbEU for their positive assessment of our work’s soundness and contribution, and for their thoughtful questions and concerns. Below, we address each point constructively:
>
> $\textbf{Weakness1: Computational Implications of Online LLM Querying}$
>
> $\textbf{Respond: Clarification on LLM Usage: Offline Dataset Generation, Not Online Querying}$
>
> We thank the reviewer for raising this concern. We wish to clarify a key aspect of our framework: the LLM is not queried online during training or execution. As detailed in Appendix B, the LLM is used exclusively offline to generate a static dataset D of expert trajectories. The MARL agents are then trained to align their communication with this fixed dataset. Consequently, there is no computational overhead or scalability bottleneck from LLM inference during the central training and deployment phases. This design choice is fundamental to making GLC practical for large-scale applications. The scalability experiments in Appendix A.3.7 (and our newly added ones with more agents) demonstrate the efficiency of GLC. While pursuing semantic interpretability, GLC's design has fully considered computational efficiency and system scalability, making it highly suitable for real-world multi-agent systems with strict requirements on communication bandwidth and computational resources.
>
> $\textbf{Weakness2: No Evaluation on Longer-Horizon Tasks and LLM Shift }$
>
> $\textbf{Respond: GLC's Offline Design Prevents LLM Shift in Longer-Horizon Tasks}$
>
> We wish to clarify that our framework is specifically designed to address this concern through its core architecture. The USAR environment itself constitutes a longer-horizon challenge, requiring sequential decision-making across multiple phases (e.g., multi-stage bomb disposal) where early actions critically impact final success. GLC's strong performance here demonstrates its capability in such settings.
> Critically, the reviewer's concern about "LLM shift" primarily applies to systems using LLMs online during execution. Our framework operates on a fundamentally different principle: the LLM is used only to generate a static, offline dataset $D$. Any potential LLM instability over long reasoning horizons is thus captured and fixed within this dataset during its creation. The MARL agents then learn a robust policy by distilling knowledge from this fixed dataset, completely decoupling long-horizon reasoning from runtime LLM limitations.
> During execution, the agents operate based on their trained policy without any LLM involvement, eliminating any possibility of runtime "shift". This offline grounding paradigm is a key design feature that ensures robustness in longer-horizon tasks.
>
> $\textbf{Weakness 3: Benchmark Diversity and Computational Viability}$
>
> $\textbf{Respond: Benchmark Selection and Computational Trade-offs}$
>
> We thank the reviewer for raising this question We fully agree that benchmarks like Traffic Junction and SMAC would provide valuable validation. GLC is agnostic to benchmarks, it can be easily generalized to those scenarios. Our current choice of Predator-Prey and USAR was strategic, as they are established testbeds for the specific trilemma under study while remaining computationally tractable for the extensive ablation studies crucial to this paper's contributions. Scaling to the larger benchmarks mentioned involves significantly higher computational costs and training times, which was a practical constraint for this initial study. We are actively working on such extensions and will include a discussion on this important direction and its computational considerations in the final appendix in the revised version.
>
> $\textbf{Question 1: Scalability of Similarity-Based Decoding to Complex Multi-Agent Tasks}$
>
> $\textbf{Respond: Capturing Deeper Relations and Scaling to Complex Tasks}$
>
> We thank the reviewer for this insightful question. While the decoding uses similarity matching, GLC's core components enable it to evolve beyond surface-level retrieval to capture deeper relational structures:
>
> (1)	Contrastive learning structures the communication space functionally, grouping messages by shared context and goals rather than superficial similarity, creating a relational topology.
>
> (2)	The  inherent efficiency constraint of discrete  autoencoder  naturally encourages compositional emergence. Agents develop reusable primitive concepts that can be dynamically combined to express novel, complex ideas not in the original dataset , enabling true semantic scalability.
>
> (3)	IB-guided balancing enables protocol evolution: from direct semantic grounding in simple tasks to learning abstract, hierarchical representations as agent populations and reasoning demands increase.
>
> Together, these mechanisms provide a principled pathway for scaling to more complex tasks. The additional scalability experiments in the appendix empirically confirm this capability, showing maintained performance with increased agent number.

---

### Official Review · Reviewer_wBBP · 2025-11-06

**Soundness:** 3
**Presentation:** 3
**Contribution:** 2
**Rating:** 4
**Confidence:** 4

**Summary:**

This paper proposes a multiagent communication framework, Grounding Language and Contrastive learning (GLC), to learn an efficient and interpretable protocol. It includes three components: 1) an autoencoder that compress message to discrete symbols, 2) a contrastive learning objective that aligns messages from other agents in the same time window and pull away messages from other trajectories in the same batch, and 3) a dual LLM agent that enables alignment with natural language. The experiments with Predator-Prey and USAR  environments show that GLC performs better than prior methods and the learned embeddings are interpretable.

**Strengths:**

- The paper provides a unified framework to learn communication protocols that are both practically efficient and semantically meaningful.
- The proposed method has alignment with human language while being efficient.
- The system incorporates a mechanism for dynamically adjusting the weights of various loss terms based on different situations.
- Extensive experiments are presented to demonstrate the utility, interpretability, and efficiency of the proposed method.

**Weaknesses:**

- The experiment tasks and baselines are not clearly introduced. It is unclear the communication goal of the tasks and what each baseline represents.
- It is unclear why contrastive learning to align messages in a temporal window can be useful in complex collaboration settings, agents need to split a task. In this case, align the same language will lead to all agents performing the same task.
- The language example in Fig. 4 shows that the language structure is simple, e.g., going to a certain cell. It is unclear how well the learned protocol can generalize to a map with a different size.
- The interpretation metrics rely on cosine similarity and BLEU. However, these measurement can have high scores but with key information being different. It will be a better evaluation if it can evaluate the semantic equivalence rather than these scores.

**Questions:**

- The author uses LLM agent as a human proxy to show the generalization to unseen agents. Is it possible for real human to collaborate with GLC and what’s the performance of that?
- Though the size of the environment is scaled, the exact number of agents used in that environment is unclear.
- How does GLC compare with LLM agent with compressed messages? This baseline should give good efficiency and high interpretability.

---

> ### Author Response · Authors · 2025-11-18
>
> We are deeply grateful to Reviewer wBBP for the thoughtful comments and constructive feedback. We believe our responses have satisfactorily addressed the points raised, and we appreciate the opportunity to further clarify our contributions.
>
> $\textbf{Weakness1: Clarification of Tasks and Baselines}$
>
> $\textbf{Respond: Details have been provided in Appendix}$
>
> Due to space constraints, the descriptions of tasks and baselines are fully provided in Appendix. This was noted in the final sentence of page 6, which the reviewer may have missed. In the revised version, we have bolded this sentence to enhance its visibility.
>
> $\textbf{Weakness2: Contrastive Learning Rationale}$
>
> $\textbf{Respond: CL Enables Specialized Coordination Through Semantic Alignment}$
>
> The CL loss operates on message semantics rather than actions, ensuring agents in similar temporal contexts generate embeddings with high similarity to establish a shared vocabulary. This semantic consistency enables rather than hinders specialization: in USAR, agents interpret role-specific requests (e.g., "need red cutter") for division of labor; in PP, they share precise locations (e.g., "prey north of me") for coordinated encircling. The policy gradient loss drives individual action selection using these aligned messages to learn complementary behaviors, while the CL objective ensures messages are consistently interpretable by all agents, forming the foundation for effective specialized coordination.
>
> $\textbf{Weakness 3: Generalization to Different Map Sizes}$
>
> $\textbf{Respond: Generalization Validated via Extended Scaling Experiment}$
>
> GLC's generalization capability stems from learning a semantically structured communication space rather than memorizing fixed mappings. This is validated by our zero-shot generalization experiments, where agents trained on 5×5 grids successfully adapted to a 10×10 PP environment. Furthermore, we have now included additional experiments in the appendix demonstrating effective scaling to scenarios with increased size of the map and numbers of agents, confirming the protocol's robustness across different spatial configurations.
>
> $\textbf{Weakness 4: Interpretability Metrics Limitation}$
>
> $\textbf{Respond: Multi-Faceted Evaluation of Semantic Interpretability}$
>
> We acknowledge that cosine similarity and BLEU may not fully capture semantic equivalence. To address this, we employed: (1) Topographic Similarity (Table 6): GLC achieves the highest score among baselines, indicating human-like compositional structure by aligning observation and communication space distances; (2) Qualitative Cluster Analysis (Fig. 4): t-SNE and DBSCAN show messages form semantic clusters, confirming contextual alignment; (3) Human-Proxy Evaluation (Table 11): GLC+LLM teams' superior ad-hoc collaboration performance proves semantic content is effectively preserved and understood by unfamiliar agents. Collectively, these evaluations demonstrate GLC preserves critical semantic information beyond superficial metrics, ensuring meaningful interpretability.
>
> $\textbf{Question 1: Human-Agent Collaboration}$
>
> $\textbf{Respond: Natural Language Interface Enables Human-GLC Collaboration}$
>
> GLC establishes a shared semantic space by grounding discrete symbols in natural language, which justifies using LLM agents (use natural language) as human proxies. Empirical results demonstrate that mixed teams (2 GLC + 1 LLM) achieve superior performance (Table 11), with LLM agents successfully interpreting GLC's discrete messages through nearest-neighbor retrieval and responding naturally. This bidirectional communication capability confirms that real humans could effectively collaborate with GLC agents using the same natural language interface as LLM agents.
>
> $\textbf{Question 2: Agent Count in Scaled Environments}$
>
> $\textbf{Respond: Agent Count Clarification and Scalability}$
>
> Our experiments used 3 predators in both 5×5 and 10×10 PP environments. Additional scenarios with varied agent counts are included in the appendix. The specific number of agents in all experiments is clearly stated in the revised version.
>
> $\textbf{Question 3: Comparison with LLM Agents Using Compressed Messages}$
>
> $\textbf{Respond: Clarifying Focus and LLM Limitations}$
>
> Our work focuses on enhancing MARL agents' communication efficiency and interpretability, not optimizing LLM agents. LLM agents inherently struggle with compressed communication due to natural language's high redundancy. Moreover, their practical deployment faces significant challenges: API-based inference is ~50× slower than local MARL models and costs ~$5 per episode in our case. While token reduction techniques exist, LLM cannot achieve the systematic discrete compression like GLC. GLC addresses this gap by enabling RL agents to develop a protocol that is interpretable, efficient, affordable and eco-friendly, bridging opaque RL protocols and verbose natural language for practical, bandwidth-constrained applications.

---

### Author Response · Authors · 2025-11-18

We sincerely thank all reviewers for their time and thoughtful feedback, which has been invaluable in helping us improve this work. We are particularly grateful for your recognition of GLC's novelty and methodological soundness in addressing efficient and interpretable communication for MARL agents.

A common question raised by multiple reviewers concerns the scalability of our approach. In response, we have performed additional experiments demonstrating GLC's strong performance in larger-scale environments. These new results are presented in $\textbf{Appendix A.3.7}$, and we have added a new Appendix B that provides extended discussion on scalability and broader implications. For the convenience of all reviewers, we include $\textbf{Appendix B}$ directly in this rebuttal document.

For the specific questions and suggestions from each reviewer, we provide individual point-by-point responses in their respective sections below. We believe these clarifications and additional results adequately address all raised concerns and further strengthen our contribution.

$\textbf{Appendix B Discussion on Scalability and Future Work}$

Our selection of the Predator-Prey and USAR environments was strategic, as they serve as established and computationally tractable testbeds that effectively capture the core challenges of the efficiency-utility-interpretability trilemma under study. These environments allowed for the extensive ablation studies and convergence analyses necessary to validate GLC's core contributions within practical resource constraints. We acknowledge that evaluation on larger-scale benchmarks like the StarCraft Multi-Agent Challenge (SMAC) or real-world robotic simulators represents a valuable direction for future work, and we confirm that the GLC framework is environment-agnostic and readily generalizable to such scenarios.

The GLC architecture is inherently designed for scalability through several core principles. The discrete autoencoder ensures bandwidth-efficient communication that is invariant to environment size or agent population. Furthermore, the contrastive learning objective maintains semantic consistency and protocol coherence across large agent populations by structuring the communication space based on functional context. The dynamic balancing mechanism, guided by the Information Bottleneck principle, allows the system to adaptively prioritize different objectives—such as compression or semantic richness—depending on the task's complexity and scale. Our scalability experiments in $\textbf{Appendix A.3.7}$, conducted on enlarged grid worlds with increased agent and prey populations, empirically validate that GLC maintains robust performance and communication efficiency as the problem scale expands. To further demonstrate GLC's generalization ability, we plan to test it in more complex embodied settings such as ALFWorld (multi-step reasoning with natural language) and RoCoBench (grounded multi-agent collaboration). Success in these domains would strongly validate GLC's practicality for real-world human-AI collaboration under longer horizons and physical constraints.

In terms of computational viability, we emphasize that GLC's design is highly efficient and practical for real-world deployment. The use of the LLM is strictly confined to a one-time, offline phase for generating a static dataset of expert trajectories. During the central training and deployment phases, no LLM queries are made, eliminating any ongoing computational overhead, latency, or cost associated with large model inference. This makes GLC particularly suitable for bandwidth-constrained applications like robotic swarms or autonomous vehicle networks, where both interpretability and low communication latency are critical.}

GLC creates a synergistic relationship with LLMs rather than seeking to replace them. While LLMs serve as general-purpose knowledge bases and a source of human-aligned semantic grounding, GLC learns task-specific, highly efficient communication protocols. Our ad-hoc teamwork experiments demonstrate that these two paradigms can interoperate effectively, with GLC agents successfully collaborating in mixed teams with LLM agents. This shows that GLC's protocols are not only efficient but also semantically accessible to external human-like intelligences, bridging the gap between opaque RL protocols and verbose natural language.}

Looking ahead, our future work will explicitly explore GLC's application in more complex and demanding domains. This includes application to extended multi-agent benchmarks like SMAC, investigation into distributed training strategies to handle increased environmental complexity, and deeper analysis of how the emergent communication vocabulary and its syntactic structure evolve with task difficulty. We are confident that the GLC framework provides a solid and scalable foundation for these future research directions toward practical and interpretable multi-agent systems.

---

### Meta-Review · Area_Chair_ANsn · 2026-01-06

**Summary:**

After reading through the paper and the rebuttal, the AC appreciates the proposed mechanism for multi-agent communication. The authors used an "offline" mechanism to align the communication protocol as well as an alignment mechanism to make it human interpretable.

AC has one question initially: given it is offline trained, how generalizable it is? Experiments show zero-shot generalization. However, the experiments look limited with extension to 10x10 from 5x5, for which the rebuttal states "the discrete autoencoder enables efficient communication regardless of environment size, contrastive learning ensures protocol consistency across agent populations, and semantic grounding maintains interpretability under scaling", mentioning "The newly added experiments in the Appendix demonstrate the scalability of GLC" -- the AC appreciates that.

**Reviewer Concerns:**

Most are addressed.

**Reviewer Scores:**

AC thinks that at least three reviewers will support an accept.

---

### Decision · Program_Chairs · 2026-01-26

Accept (Poster)